# Instance-Specific Asymmetric Sensitivity in Differential Privacy

**David Durfee**
Mozilla Anonym
ddurfee@mozilla.com

## Abstract

We provide a new algorithmic framework for differentially private estimation of general functions that adapts to the hardness of the underlying dataset. We build upon previous work that gives a paradigm for selecting an output through the exponential mechanism based upon closeness of the inverse to the underlying dataset, termed the inverse sensitivity mechanism. Our framework will slightly modify the closeness metric and instead give a simple and efficient application of the sparse vector technique. While the inverse sensitivity mechanism was shown to be instance optimal, it was only with respect to a class of unbiased mechanisms such that the most likely outcome matches the underlying data. We break this assumption in order to more naturally navigate the bias-variance tradeoff, which will also critically allow for extending our method to unbounded data. In consideration of this tradeoff, we provide theoretical guarantees and empirical validation that our technique will be particularly effective when the distances to the underlying dataset are asymmetric. This asymmetry is inherent to a range of important problems including fundamental statistics such as variance, as well as commonly used machine learning performance metrics for both classification and regression tasks. We efficiently instantiate our method in $O(n)$ time for these problems and empirically show that our techniques will give substantially improved differentially private estimations.

## 1  Introduction

We consider the general problem of estimating aggregate functions or statistics of a dataset with differential privacy. The massive increase in data collection to improve analytics and modelling across industries has made such data computations invaluable, but can also leak sensitive individual information. Rigorously measuring such leakage can be achieved through differential privacy, which quantifies the extent that one individual's data can affect the output. Much of the focus within the field of differential privacy is upon constructing algorithms that give both accurate output and privacy guarantees by injecting specific types of randomness. One of the most canonical mechanisms for achieving this considers the maximum effect one individual's data could have upon the output of a given function, referred to as the *sensitivity* of the function, and adds proportional noise to the function output. In general, the notion of sensitivity plays a central role in many differentially private algorithms, directly affecting the accuracy of the output.

While using the worst-case sensitivity across all potential datasets will ensure privacy guarantees, the utility can be improved by using variants of sensitivity that are specific to the underlying dataset. This notion was initially considered in  Nissim et al. (2007), introducing *smooth sensitivity*, an interpolation between worst-case sensitivity and *local sensitivity* of the underlying data, by which noise could be added proportionally. The smooth sensitivity adapts well to the underlying data and

was further extended to other commonly used variants of the original privacy definition Bun & Steinke (2019).

More aggressive methods were also considered with a data-independent conjectured sensitivity parameter and more accurate results provided when the underlying data complies with the parameter. The propose-test-release methods check that all datasets close to the underlying data have sensitivity below a parameter and add noise proportional when the criteria is met and fail otherwise Dwork & Lei (2009); Thakurta & Smith (2013). Preprocessing methods provide an approximation of the function with sensitivity below a given parameter by which noise can be added proportionally and the approximation is accurate for underlying data with low sensitivity Chen & Zhou (2013); Blocki et al. (2013); Kasiviswanathan et al. (2013); Cummings & Durfee (2020). Clipping techniques, commonly seen in differentially private stochastic gradient descent Abadi et al. (2016), are also a more rudimentary and efficient preprocessing method for ensuring sufficiently small sensitivity. The primary challenge with these approaches is the sensitivity parameter must be specified *a priori* and can add significant bias if the underlying data does not comply with the parameter.

In contrast, the inverse sensitivity mechanism directly improves upon the smooth sensitivity technique adapting even better to the underlying data. While several instantiations had been previously known in the literature, it was introduced in it's full generality in Asi & Duchi (2020b). Specifically, this framework considers all potential outputs based upon the closeness of their inverse to the underlying data and applies the exponential mechanism to select a point accordingly. This exact methodology can even improve upon adding noise proportional to the local sensitivity of the underlying data, which generally violates differential privacy. Follow-up work gave approximations of this method that allow for efficient implementations of more complex instantiations Asi & Duchi (2020a). For both the exact and approximate versions, the inverse sensitivity mechanism is instance optimal and nearly instance optimal, respectively, under certain assumptions Asi & Duchi (2020a,b).

## 1.1 Our techniques

We build upon the inverse sensitivity mechanism, particularly within the class of functions for which it was shown to be optimal. However, those guarantees only held for a class of unbiased mechanisms such that the most likely outcome matches the underlying data. The inverse sensitivity mechanism and smooth sensitivity techniques fit this characterization of unbiased. The methods that specify a data-independent sensitivity parameter break this assumption by essentially fixing the variance through this parameter but adding significant bias if the variance parameter is set too low. Our method will also break the unbiased assumption but still adapt well to the underlying data to more naturally navigate the bias-variance tradeoff.

In particular, we similarly consider the distance from the underlying data to the inverse of each possible output, which can be considered the inverse sensitivity. However, we instead invoke the well-known sparse vector technique, originally introduced in Dwork et al. (2009), to select an output close to the underlying data. The iterative nature of sparse vector technique will create a slight bias, while still adapting well to the underlying data. By utilizing this iterative technique, we can also better take advantage when the sensitivities are asymmetric that allows us to reduce the variance, and we thusly term our method the *asymmetric sensitivity mechanism*. In fact, the local sensitivity can be infinite with unbounded data and our technique can still naturally handle this setting for a wide variety of functions including our instantiations. We support this with theoretical utility guarantees that are asymptotically superior to previous work under these conditions.

Our notion of asymmetric sensitivities is inherent to a range of problems, and we first instantiate our method upon variance, a fundamental property of a dataset that is widely used in statistical analysis. Likewise, this property will also apply to commonly used machine learning performance metrics: cross-entropy loss, mean squared error (MSE), and mean absolute error (MAE). Model performance evaluation is an essential part of a machine learning pipeline, particularly for iterative improvement, so accurate and private evaluation is critical. We instantiate our method upon these functions as well, and give an extensive empirical study for each instantiation across a variety of datasets and privacy parameters. We show that our method significantly improves performance of private estimation for these important problems. We further complement our results with an approximate method that allows for more efficient implementations of general functions while still preserving the asymmetry that we exploit for improved estimations. This will allow us to give $O(n)$ time implementations for each invocation of our method.

## 1.2 Additional related works

While the most closely related literature was discussed in more detail previously, we provide additional related works here. Recent work considered instance-optimality but for estimators population quantities McMillan et al. (2022), which differ from the empirical quantities studied here and in the other previously mentioned work. Additional work formally considered the bias-variance-privacy trade-off particularly for mean estimation Kamath et al. (2023), but considers bias in the more classical sense. Interestingly, it's also been seen in the work for obtaining (asymptotically) optimal mean estimation for subgaussian distributions Karwa & Vadhan (2018); Bun & Steinke (2019) and distributions satisfying bounded moment conditions Barber & Duchi (2014); Kamath et al. (2020), that adding bias was necessary. This fits with our results where bias, albeit a different type, was needed to improve instance-specific differential privacy.

## 1.3 Our contributions

We summarize our primary contributions as the following:

1. We introduce a new algorithmic framework for private estimation of general functions, which we refer to as the asymmetric sensitivity mechanism, along with a more computationally efficient approximate variant (see Section 3).

2. We provide theoretical utility guarantees that asymptotically confirm our method's advantage when the sensitivities are asymmetric and further give intuition and empirical support of this asymmetric advantage (see Section 4)

3. We efficiently instantiate our method for private variance estimation, and provide an extensive empirical study showing significantly improved accuracy (see Section 5).

4. We further invoke our method upon model evaluation for both classification and regression tasks with corresponding efficient implementations and empirical studies showing improved estimations (see Section 6).

Additional and supplemental analysis, results and empirical studies are pushed to the appendix.

## 2 Preliminaries

For simplicity and ease of comparison we borrow much of the notation from Asi & Duchi (2020a,b).

**Definition 2.1.** Let $x, x'$ be datasets of our data universe $\mathcal{X}^n$. We define $d_{\mathsf{ham}}(x, x') = |\{i : x_i \neq x'_i\}|$ to be the Hamming distance between datasets. If $d_{\mathsf{ham}}(x, x') \leq 1$ then $x, x'$ are *neighboring* datasets.

Note that we assume the *swap* definition of neighboring datasets but will also discuss how our results apply to the *add-subtract* definition in Appendix B.3. We further define the (global) sensitivity.

**Definition 2.2.** $f : \mathcal{X}^n \to \mathbb{R}$ has sensitivity $\Delta$ if for any neighboring datasets $|f(x) - f(x')| \leq \Delta$

We will be using the classical (pure) differential privacy definition, but will also discuss how our methods apply to other definitions with improved guarantees.

**Definition 2.3.** Dwork et al. (2006b,a) A mechanism $M : \mathcal{X}^n \to \mathcal{T}$ is $(\varepsilon, \delta)$-differentially-private (DP) if for any neighboring datasets $x, x' \in \mathcal{X}$ and measurable $S \subseteq \mathcal{T}$:

$$\Pr[M(x) \in S] \leqslant e^\varepsilon \Pr[M(x') \in S] + \delta.$$

If $\delta = 0$ then $M$ is $\varepsilon$-DP.

### 2.1 Sparse vector technique

We define the fundamental sparse vector technique introduced in Dwork et al. (2009) and often considered to apply Laplacian noise Lyu et al. (2017). However, recent work showed the noise can instead be added from the exponential distribution at the same parameter for improved utility Durfee (2023). Let $\mathsf{Expo}(b)$ denote a draw from the exponential distribution with scale parameter $b$. The sparse vector technique iteratively calls the following algorithm.

This technique can further see improvement when the queries are monotonic which will apply to most of our instantiations of our method.

---

**Algorithm 1** `AboveThreshold`

---

**Require:** Input dataset $\boldsymbol{x}$, a stream of queries $\{f_i \,:\, \mathcal{X}^n \to \mathbb{R}\}$ with sensitivity $\Delta$, and a threshold $T$

1: Set $\hat{T} = T + \text{Expo}(\Delta/\varepsilon_1)$
2: **for** each query $i$ **do**
3:     Set $v_i = \text{Expo}(\Delta/\varepsilon_2)$
4:     **if** $f_i(\boldsymbol{x}) + v_i \geq \hat{T}$ **then**
5:         Output $\top$ and `halt`
6:     **else**
7:         Output $\bot$
8:     **end if**
9: **end for**

---

**Definition 2.4.** We say that stream of queries $\{f_i \,:\, \mathcal{X}^n \to \mathbb{R}\}$ with sensitivity $\Delta$ is *monotonic* if for any neighboring $\boldsymbol{x}, \boldsymbol{x}' \in \mathcal{X}^n$ we have either $f_i(\boldsymbol{x}) \leq f_i(\boldsymbol{x}')$ for all $i$ or $f_i(\boldsymbol{x}) \geq f_i(\boldsymbol{x}')$ for all $i$.

This allows for the following differential privacy guarantees from Durfee (2023).

**Proposition 2.5.** *Algorithm 1 is $(\varepsilon_1 + 2\varepsilon_2)$-DP in general and $(\varepsilon_1 + \varepsilon_2)$-DP for monotonic queries*

## 2.2 Inverse sensitivity mechanism

The inverse sensitivity mechanism had seen several previous instantiations but was introduced in it's full generality in Asi & Duchi (2020b). We first introduce the exponential mechanism.

**Definition 2.6.** McSherry & Talwar (2007) The Exponential Mechanism is a randomized mapping $M \,:\, \mathcal{X}^n \to \mathcal{T}$ such that

$$\Pr\left[M(\boldsymbol{x}) = t\right] \propto \exp\left(\frac{\varepsilon \cdot q(\boldsymbol{x}, t)}{2\Delta}\right)$$

where $q \,:\, \mathcal{X}^n \times \mathcal{T} \to \mathbb{R}$ has sensitivity $\Delta$.

**Proposition 2.7.** *McSherry & Talwar (2007) The exponential mechanism is $\varepsilon$-DP*

We then define the distance of a potential output from the underlying dataset to be the Hamming distance required to change the data such that the new data matches the output.

**Definition 2.8.** For a function $f \,:\, \mathcal{X}^n \to \mathcal{T}$ and $\boldsymbol{x} \in \mathcal{X}^n$, let the *inverse sensitivity* of $t \in \mathcal{T}$ be

$$\text{len}_f(\boldsymbol{x}; t) \overset{\text{def}}{=} \inf_{\boldsymbol{x}'}\{d_{\text{ham}}(\boldsymbol{x}, \boldsymbol{x}') | f(\boldsymbol{x}') = t\}$$

By construction this distance metric for any output cannot change by more than one between neighboring datasets due to the triangle inequality for Hamming distance.

**Corollary 2.9.** *For any neighboring datasets $\boldsymbol{x}, \boldsymbol{x}' \in \mathcal{X}^n$ and $t \in \text{im}(f)$ where $\text{im}(f) \subseteq \mathcal{T}$ is the image of the function, we have $|\text{len}_f(\boldsymbol{x}; t) - \text{len}_f(\boldsymbol{x}'; t)| \leq 1$*

The *inverse sensitivity mechanism* then draws from the exponential mechanism instantiated upon the distance metric giving the density function

$$\pi_{M_{\text{inv}}(\boldsymbol{x})}(t) = \frac{e^{-\text{len}_f(\boldsymbol{x}; t)\varepsilon/2}}{\int_{\mathcal{T}} e^{-\text{len}_f(\boldsymbol{x}; s)\varepsilon/2}ds} \tag{M.1}$$

and mechanism M.1 is $\varepsilon$-DP by Proposition 2.7 and Corollary 2.9.

# 3 Asymmetric Sensitivity Mechanism

In this section we introduce our general methodology for instance-specific differentially private estimation, which we term the asymmetric sensitivity mechanism. We first give the exact formulation which will fit an extensive class of functions focused upon in Asi & Duchi (2020b). Next

we provide a simple framework by which our method can be implemented. The efficiency of this implementation is highly dependent upon the function of interest, but we supplement these results with an approximate method. This can allow for broader efficient implementations and also extends our methodology to general functions. While this section will set up and provide the necessary rigor for our techniques, we also point the reader to Appendix C for a more intuitive explanation of our approach compared to the inverse sensitivity mechanism.

## 3.1 Exact asymmetric sensitivity mechanism

Our method will similarly consider the distance for each output from the underlying data with the goal being to select an output with distance close to zero. The inverse sensitivity mechanism does this through the exponential mechanism, but we will instead apply the sparse vector technique. In order to better apply the sparse vector technique, we will first modify the inverse sensitivity such that it is negative for outputs that are less than output from the underlying data.

**Definition 3.1.** For $f : \mathcal{X}^n \to \mathbb{R}$ and $x \in \mathcal{X}^n$, let the *reflective inverse sensitivity* of $t \in \mathbb{R}$ be

$$\mathsf{s}_f(x;t) \overset{\text{def}}{=} \text{sgn}(t - f(x)) \left( \text{len}_f(x;t) - \frac{1}{2} \right)$$

Intuitively, the goal of applying the sparse vector technique will be to identify when the reflective inverse sensitivity crosses the threshold from negative to positive. While there are reasonable methods of extending our approach to higher dimensions, it will both become computationally inefficient and the notion of asymmetry, which gives our method the most significant improvement, is less inherent in higher dimensions. Initially, we focus upon a general class of functions considered in Asi & Duchi (2020b) that was shown to include all continuous functions from a convex domain.

**Definition 3.2** (Definition 4.1 in Asi & Duchi (2020b)). Let $f : \mathcal{X}^n \to \mathbb{R}$. Then $f$ is *sample-monotone* if for every $x \in \mathcal{X}^n$ and $s, t \in \mathbb{R}$ satisfying $f(x) \le s \le t$ or $t \le s \le f(x)$, we have $\text{len}_f(x;s) \le \text{len}_f(x;t)$

For this class of functions, we show that the reflective inverse sensitivity of an output maintains closeness between neighboring datasets. Accordingly, we can apply the sparse vector technique to a stream of potential outputs in order to (noisily) identify when the reflective inverse sensitivity crosses the threshold from negative to positive. This gives the *asymmetric sensitivity mechanism* for a stream of potential outputs $\{t_i\}$ that calls AboveThreshold and returns $t_k$ when

$$\text{AboveThreshold}(x, \{\mathsf{s}_f(x;t_i)\}, T = 0) = \{\perp^{k-1}, \top\} \tag{M.2}$$

To be effective, this stream of potential outputs should be increasing (or decreasing if we flip the sign of $\mathsf{s}_f$) but will still achieve the desired privacy guarantees regardless which is shown in Appendix A.2.

**Theorem 3.3.** *Given sample-monotone $f : \mathcal{X}^n \to \mathbb{R}$ and any stream of potential outputs $\{t_i\}$, we have that mechanism M.2 is $(\varepsilon_1 + 2\varepsilon_2)$-DP in general and $(\varepsilon_1 + \varepsilon_2)$-DP if $\mathsf{s}_f$ is monotonic.*

We further detail in Appendix B a simple, general, and robust strategy for selecting potential outputs that provides a reasonable limit on the number of queries.

## 3.2 Implementation framework

The primary bottleneck in efficiently implementing both our asymmetric sensitivity mechanism and the inverse sensitivity mechanism is the computation of the inverse sensitivity. In particular, it will require computing upper and lower output bounds for different Hamming distances from our underlying data.

**Definition 3.4.** For a function $f : \mathcal{X}^n \to \mathbb{R}$, we define the upper and lower output bounds for Hamming distance $\ell$ as

$$U_f^\ell(x) \overset{\text{def}}{=} \sup_{x'}\{f(x') : d_{\text{ham}}(x, x') \le \ell\}$$

and

$$L_f^\ell(x) \overset{\text{def}}{=} \inf_{x'}\{f(x') : d_{\text{ham}}(x, x') \le \ell\}$$

The complexity of computing these depends upon the function, but we can use the upper and lower output bounds to get the inverse sensitivity with the following lemma proven in Appendix A.2.

**Lemma 3.5.** *If $f$ is sample-monotone then $\mathrm{len}_f(\boldsymbol{x};t) = \inf\{\ell : L_f^\ell(\boldsymbol{x}) \le t \le U_f^\ell(\boldsymbol{x})\}$ for all $t \in \mathbb{R}$*

If we then assume access to the array $L_f^n(\boldsymbol{x}), ..., L_f^1(\boldsymbol{x}), f(\boldsymbol{x}), U_f^1(\boldsymbol{x}), ..., U_f^n(\boldsymbol{x})$, for any potential output $t_i$ we can compute $\mathrm{len}_f(\boldsymbol{x};t_i)$ in $O(\log(n))$ time with a simple binary search. Alternatively, we could also take $O(n)$ amortized time by maintaining a pointer and iteratively increasing the index for each new potential output if we assume the stream of potential outputs are non-decreasing. This gives the general implementation framework:

1. Compute upper and lower output bounds $U_f^\ell(\boldsymbol{x})$ and $L_f^\ell(\boldsymbol{x})$ for all $\ell \in [n]$

2. Use the output bounds to efficiently run $\mathrm{AboveThreshold}(\boldsymbol{x}, \{\mathsf{s}_f(\boldsymbol{x};t_i)\}, T = 0)$

### 3.3 Approximate asymmetric sensitivity mechanism

In Section A, we show how we can extend our asymmetric sensitivity mechanism to general functions $f : \mathcal{X}^n \to \mathbb{R}$ and provide more efficient implementations. It will follow closely with our exact version above.

## 4 Asymmetric Sensitivity Advantage

In this section, we first connect our definitions with the corresponding definitions in the previous work, by which utility guarantees are provided. Then we discuss the notion of asymmetric sensitivities and provide our utility guarantees that exploit this asymmetry to asymptotically improve upon the previous work under those conditions.

### 4.1 Connection to previous work

An essential quantity for our method and both inverse and smooth sensitivity mechanisms is the amount a function output can change if $k$ individuals change their data. This is quantified in Equation 3 from Asi & Duchi (2020b) which can be translated to our definitions (in the case when $\mathcal{T} = \mathbb{R}$) as

$$\omega_f(\boldsymbol{x};k) \overset{\text{def}}{=} \max\{|f(\boldsymbol{x}) - L_f^k(\boldsymbol{x})|, |f(\boldsymbol{x}) - U_f^k(\boldsymbol{x})|\}$$

Note that if $k = 1$ then this is the *local sensitivity* of the function. It's then shown in Asi & Duchi (2020b) (Corollary 4.2 and Equation 13, respectively) that the general utility guarantees of both inverse sensitivity mechanism and smooth sensitivity mechanism are bounded with respect to this quantity. More simply, the accuracy guarantees degrade as the local sensitivity increases and there is no utility bound if local sensitivity is infinite.

### 4.2 Asymmetric accuracy guarantees

To understand the advantages of our method, we will consider the sensitivities to be asymmetric if $|f(\boldsymbol{x}) - U_f^k(\boldsymbol{x})| \gg |f(\boldsymbol{x}) - L_f^k(\boldsymbol{x})|$ for most $k$ (or vice versa), which is to say that changing an individual's data can generally increase the function more than decrease it. In general, we expect any lower bounded function to inherently limit the amount changing one individual's data can decrease the function compared to increasing the function. Each instantiation in our empirical study is a non-negative function which then fits this characterization. For simplicity, we will restrict our consideration to non-negative functions for our utility guarantees, but can easily apply these to other settings.

The goal for our method is to exploit the asymmetric sensitivities by instead applying the sparse vector technique. The iterative nature of this technique biases the output towards being less than $f(\boldsymbol{x})$, but more importantly the $U_f^k(\boldsymbol{x})$ values will have little effect upon the accuracy. Specifically, once the threshold is crossed it becomes increasingly unlikely that the sparse vector technique will proceed. Explicitly connecting this with our mechanism, even if $U_1^k(\boldsymbol{x}) = \infty$ and so the local

sensitivity is infinite, we still have $\mathsf{s}_f(\boldsymbol{x};t_i) = 1/2$ for all $t_i > f(\boldsymbol{x})$ and assuming $t_i$ is increasing it is increasingly unlikely that we output larger $t_i > f(\boldsymbol{x})$.

We formalize this intuition by providing a theoretical utility guarantee that does not depend upon the upper bound values. Essentially, we are able to replace the $|f(\boldsymbol{x}) - U_f^k(\boldsymbol{x})|$ in $\omega_f(\boldsymbol{x};k)$ with a relative error bound based upon a parameter that we fix across all our empirical instantiations. Accordingly, as $U_f^1(\boldsymbol{x}) \to \infty$ and thereby asymmetry increases and also local sensitivity increases, our utility guarantees are asymptotically superior to both inverse and smooth sensitivity mechanisms.

**Lemma 4.1.** *Let $M$ denote the mechanism from M.2 with $t_i = \beta^i - 1$ where $\beta > 1$ and we let $\varepsilon = \varepsilon_1 + 2\varepsilon_2$ with $\varepsilon_1 = \varepsilon_2$ in the call to Algorithm 1. Given non-negative sample monotone $f : \mathcal{X}^n \to \mathbb{R}$ we have*

$$Pr\left[|M(\boldsymbol{x}) - f(\boldsymbol{x})| < \max\{|f(\boldsymbol{x}) - L^{\log k+1}(\boldsymbol{x})|, (\beta^k - 1)(f(\boldsymbol{x}) + 1)\}\right] > 1 - O\left(\frac{1}{ke^{\varepsilon/6}}\right)$$

*for any $\boldsymbol{x} \in \mathcal{X}^n$ such that $f(\boldsymbol{x}) \leq \beta^{O(1)}$.*

We provide the proof in Appendix C along with further intuition and empirical evidence of our asymmetric advantage. We also extend these results to general non-negative functions in Corollary C.4 by applying our approximate variant to achieve the same guarantees. We set $\beta = 1.005$ in all our empirical studies and also note that our method is robust to reasonable choices of the $\beta$ parameter (Appendix B.2).

## 5 Private Variance Estimation

In this section, we instantiate our asymmetric sensitivity mechanism upon variance, a fundamental property of a dataset that is widely used in statistical analysis.

**Definition 5.1.** Let $\mathcal{X} = \mathbb{R}$ and for $\boldsymbol{x} \in \mathcal{X}^n$ we let $\bar{x} = \frac{1}{n}\sum_{i=1}^n \boldsymbol{x}_i$ and define

$$\mathbf{Var}\left[\boldsymbol{x}\right] \stackrel{\text{def}}{=} \frac{1}{n}\sum_{i=1}^n (\boldsymbol{x}_i - \bar{x})^2$$

There has been extensive work in the privacy literature upon covariance estimation for univariate and multivariate Gaussians with a focus upon optimizing asymptotic performance[1]. Our focus here will be practical methods for general data, so a rigorous comparison to all these methods for Gaussians is untenable and outside the scope of this work.

We first show how the variance instantiation of asymmetric sensitivity can be implemented efficiently and give intuition upon why we expect asymmetric sensitivities for this function. Next we give a detailed empirical study that confirms this intuition, showing that our method will substantially outperform inverse sensitivity on the task of variance estimation.

### 5.1 Efficient variance instantiation

As seen in Section 3.2, we need to efficiently provide upper and lower output bounds in order to achieve an efficient implementation. We first consider the lower output bounds and can provide the exact bounds from Definition 3.4 which we prove in Appendix D.

**Lemma 5.2.** *Given $\boldsymbol{x} \in \mathbb{R}^n$, if $\boldsymbol{x}_1 \leq ... \leq \boldsymbol{x}_n$ are ordered then we have lower output bounds*

$$L_{\mathbf{Var}}^{\ell}(\boldsymbol{x}) = \frac{n - \ell}{n} \min_{0 \leq i \leq \ell} \mathbf{Var}\left[\boldsymbol{x}_{[\ell+1-i:n-i]}\right]$$

*where we let $\boldsymbol{x}_{[\ell+1-i:n-i]} \stackrel{\text{def}}{=} (\boldsymbol{x}_{\ell+1-i}, ..., \boldsymbol{x}_{n-i})$*

---

[1]See Karwa & Vadhan (2018); Du et al. (2020); Biswas et al. (2020); Kamath et al. (2019); Bun et al. (2019); Aden-Ali et al. (2021); Ashtiani & Liaw (2022); Kothari et al. (2022); Tsfadia et al. (2022); Liu et al. (2022); Kamath et al. (2022)

While the formula above immediately suggests an $O(n^2)$ time computation of all the lower output bounds, we will further prove in Appendix D that we can use our approximation to get a more efficient implementation. In particular, we consider a data independent fixed distance and only compute the exact bounds outputs closer to the underlying data to still ensure accurate estimations.

**Lemma 5.3.** *Given $x \in \mathbb{R}^n$ and $c \geq 0$, we can compute all approximate output bounds $\bar{L}^\ell_{\mathbf{Var}}(x) = L^\ell_{\mathbf{Var}}(x)$ for $\ell \leq c$ and $\bar{L}^\ell_{\mathbf{Var}}(x) = 0$ for $\ell > c$ in $O(n + c^2)$ time.*

Next we consider the upper output bounds, but if the data is unbounded then we must have $U^1_{\mathbf{Var}}(x) = \infty$. It is precisely for this reason that asymmetric sensitivities are inherent for variance. Our method can naturally handle this setting and we show in Appendix B.1 that unbounded upper output bounds barely affects our accuracy. However, applying the inverse sensitivity mechanism requires reasonable bounds upon each data point that should be data-independent and also sufficiently loose to not add bias from clipping data points.

**Lemma 5.4.** *If we restrict all values to the interval $[a, b]$ then given $x \in [a, b]^n$ we can give approximate upper output bounds*

$$\overline{U}^\ell_{\mathbf{Var}}(x) = \mathbf{Var}\left[x\right] + \frac{\ell(b - a)^2}{n}$$

To our knowledge, there is no efficient method for computing the exact upper output bounds for general data (contained in a range), so to maintain practicality we provide approximate bounds, proven in Appendix D.

---

**Algorithm 2** Variance instantiation of asymmetric sensitivity mechanism

---

**Require:** Input dataset $x$, and parameter $\beta > 1$
 1: Compute all $\bar{L}^\ell_{\mathbf{Var}}(x)$ with $c = 100$ (Lemma 5.3)
 2: Compute all $\overline{U}^\ell_{\mathbf{Var}}(x)$ if domain is restricted to $[a, b]^n$ (Lemma 5.4)
 3: $\{\perp^{k-1}, \top\} \longleftarrow \texttt{AboveThreshold}(x, \{\overline{s}_f(x; \beta^i - 1)\}, T = 0)$
**output** $\beta^k - 1$

---

**Theorem 5.5.** *Algorithm 2 is $(\varepsilon_1 + 2\varepsilon_2)$-DP and has a runtime of $O(n + q)$ where $q$ is the number of queries in* `AboveThreshold`

*Proof.* If we assume the domain is restricted to $[a, b]^n$ then the privacy guarantees follow from Lemma 5.3 and Lemma 5.4 applied to Theorem A.4. If not then we apply Lemma 5.3 and $\overline{U}^1_{\mathbf{Var}}(x) = \infty$ to Theorem A.4 to get our privacy guarantees.

For the runtime, computing all $\bar{L}^\ell_{\mathbf{Var}}(x)$ is $O(n)$ time by Lemma 5.3 and fixing $c = 100$, and computing all $\overline{U}^\ell_{\mathbf{Var}}(x)$ is $O(n)$ time. Finally, we can run `AboveThreshold` in $O(n + q)$ time as seen in Section 3.2 □

In our implementations we'll more reasonably assume $\beta \geq 1.001$, so $\beta^{50000} > 10^{21}$ and we'll simply terminate `AboveThreshold` after at most 50,000 queries for all datasets without affecting the privacy guarantees. This then gives a runtime of $O(n)$.

## 5.2 Empirical study of variance estimation

For our instantiations of machine learning model evaluation we will be using the following datasets for regression tasks: Diamonds dataset containing diamond prices and related features Wickham (2016); Abalone dataset containing age of abalone and related features Nash et al. (1995); and Bike dataset containing number of bike rentals and related features Fanaee-T (2013). We will also use the labels from these datasets to test our variance invocation. We also use the Adult dataset, Becker & Kohavi (1996), for model evaluation of classification tasks so we will borrow two of the features, age and hours worked per week, to test our variance invocation.

While our method does not require any bounds on the data to still maintain high accuracy (see Appendix B.1), it is necessary for the other mechanisms. All of our data is inherently non-negative

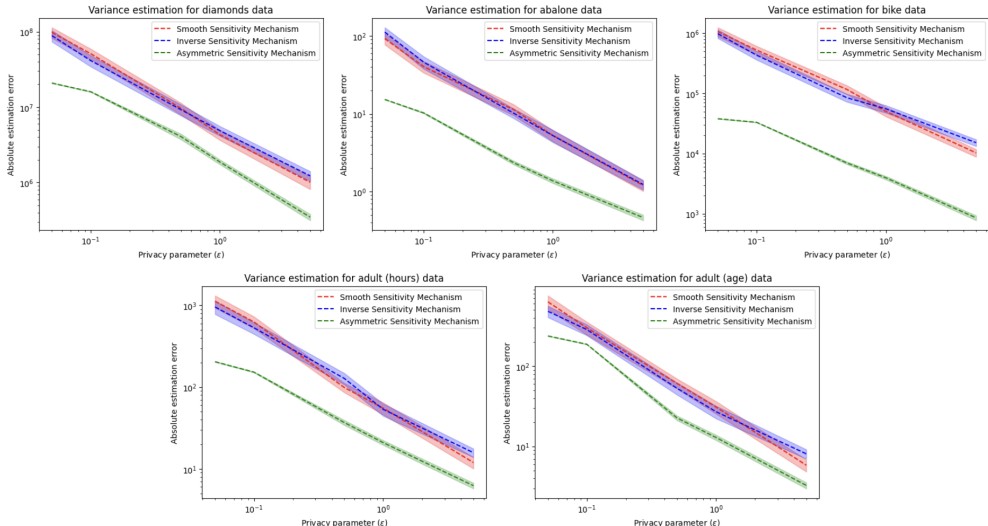

Figure 1: Plots comparing each method for estimating variance. For each privacy parameter we sample 1,000 datapoints from the dataset and call each mechanism 100 times, plotting the average absolute error with 0.9 confidence intervals.

and some data has innate upper bounds. If not, we use reasonable upper bounds, which should be data independent, to avoid adding bias from clipping the data. We use the following bounds: [0,50000] for diamond prices; [0,50] for abalone ages; [0,5000] for city bike rentals; [0,168] for hours; and [0,125] for age. Our algorithm (Algorithm 2 in Appendix D) will have a parameter $\beta$ which we fix $\beta = 1.005$ and will maintain this consistency across all experiments. We further show in Appendix B.2 that our method is robustly accurate across reasonable $\beta$ choices.

For each privacy parameter, we repeat 100 times: sample 1,000 datapoints from the dataset and call each mechanism on the sampled data for estimates. We plot the average absolute error for each method along with confidence intervals of 0.9 in Figure 5.2. As expected, we see that our approach of variance estimation sees substantially less error across privacy parameters and datasets.

## 6 Private Machine Learning Model Evaluation

In this section, we invoke our asymmetric sensitivity mechanism upon commonly used metrics for machine learning model evaluation for both classification and regression tasks. In particular, we consider cross-entropy loss for classification tasks, and mean squared error (MSE) and mean absolute error (MAE) for regression tasks. Note that combining our improved estimation for variance in Section 5 with the improved estimation for MSE also implies an improved estimation of the coefficient of determination, $R^2$, also commonly used for evaluating regression performance.

We provide full definitions along with technical analysis in Section E.

### 6.1 Empirical study of model evaluation for classification

For our empirical study of model evaluation for classification tasks we will consider two tabular datasets with binary labels, the Adult dataset Becker & Kohavi (1996) and Diabetes dataset Efron et al. (2004), along with two computer vision tasks with 10 classes, the mnist dataset LeCun et al. (2010) and cifar10 dataset Krizhevsky et al. (2009). Our focus here is upon the accuracy of our evaluation, not optimizing the quality of the model itself. As such, we will be using reasonable choices for models for simplicity but certainly not state-of-the-art models .

For the tabular data, we partition into train and test with an 80/20 split and train with an xgboost classifier with the default parameters. For the mnist data, which is already partitioned, we use a simple MLP with one inner dense layer of 128 neurons and relu activation, and the final layer of 10 neurons has a softmax activation. We train this model for 5 epochs. For the cifar10 data, which is already partitioned, we use a relatively small CNN with several pooling and convolutional layers,

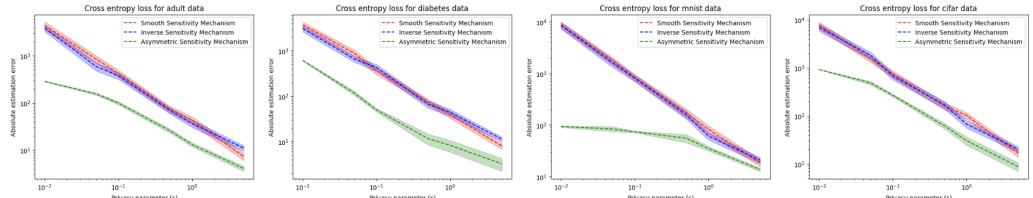

Figure 2: Plots comparing each method for estimating cross entropy loss. For each privacy parameter we both sample 1,000 datapoints from the test set and call each mechanism 100 times, plotting the average absolute error with 0.9 confidence intervals.

and several dense layers at the end with relu activation and final layer with softmax activation. We train this model for 10 epochs.

Again our method does not require any bounds on the data to maintain high accuracy, but it is necessary for the inverse sensitivity mechanism. Given the softmax activation for our models, the outputs are unbounded, but we will provide reasonable bounds. We will use the bounds [-10,10] of the model output for binary classification tabular data, and bounds $[-25, 25]^{10}$ of the model output for the multi-classification vision data. Once again, we fix our parameter $\beta = 1.005$.

## 6.2 Empirical study of model evaluation for regression

As discussed in Section 5, our machine learning model evaluations for regression will use the following datasets: Diamonds dataset containing diamond prices and related features Wickham (2016); Abalone dataset containing age of abalone and related feature Nash et al. (1995); and Bike dataset containing number of bike rentals and related feature Fanaee-T (2013). We also use the same parameters from Section 5 for these datasets. Once again, our goal here is to accurately assess the quality of the model and not optimize performance. As such we simply train with xgboost regressor under default parameters after partitioning each dataset into train and test with an 80/20 split.

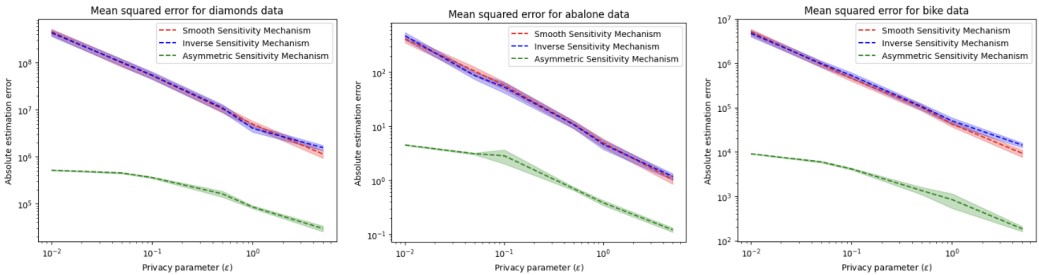

We first consider mean squared error estimation and repeat 100 times for each privacy parameter: draw 1,000 datapoints from the test set and call each mechanism 100 times for estimates. We then plot the average absolute error along with confidence intervals of 0.9. We repeat this process for mean absolute error.

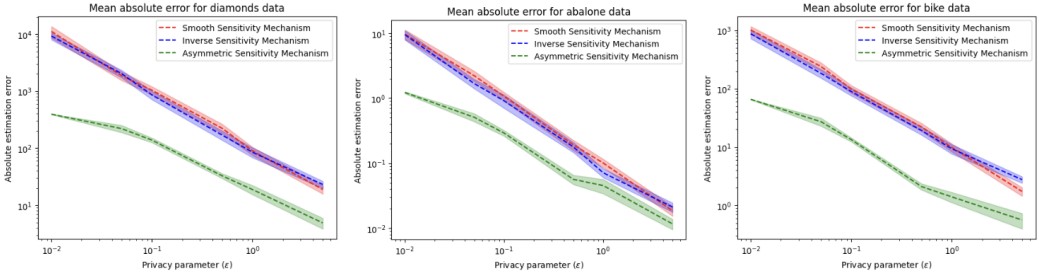

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

# A  Additional Analysis of Asymmetric Sensitivity Mechanism

In this section, we give full details upon extending our asymmetric sensitivity mechanism to general functions $f : \mathcal{X}^n \to \mathbb{R}$ and provide more efficient implementations. We also provide all missing proofs from Section 3.

## A.1  Approximate asymmetric sensitivity mechanism

As noted in Section 3, computing the upper and lower outputs bounds can be inefficient or even infeasible, which is necessary for our method, inverse sensitivity method, and smooth sensitivity method. As such we provide an approximation of these bounds that is a slightly more granular version of the approximation method provided in Asi & Duchi (2020a) and also applies to the inverse sensitivity mechanism.

**Definition A.1.** For a function $f : \mathcal{X}^n \to \mathbb{R}$, we define approximate upper and lower sensitivity bounding functions of Hamming distance $\ell$ to be $\overline{U}_f^\ell : \mathcal{X}^n \to \mathbb{R}$ and $\bar{L}_f^\ell : \mathcal{X}^n \to \mathbb{R}$, if for all $\ell \geq 0$ and any neighboring datasets $x, x' \in \mathcal{X}^n$ we have $\overline{U}_f^\ell(x) \geq U_f^\ell(x)$ and $\overline{U}_f^\ell(x) \leq \overline{U}_f^{\ell+1}(x')$ along with $\bar{L}_f^\ell(x) \leq L_f^\ell(x)$ and $\bar{L}_f^\ell(x) \geq \bar{L}_f^{\ell+1}(x')$

In particular, this definition separates the approximation for the upper vs lower bounds as opposed to treating them symmetrically. This maintains asymmetry which is precisely where our technique excels most. We then define a variant of closeness for each output to the underlying data which utilizes these approximate upper and lower bounds.

**Definition A.2.** For a function $f : \mathcal{X}^n \to \mathbb{R}$ along with $\overline{U}_f^\ell : \mathcal{X}^n \to \mathbb{R}$ and $\bar{L}_f^\ell : \mathcal{X}^n \to \mathbb{R}$, for any $t \in \mathbb{R}$, we let

$$\overline{\mathsf{len}_f}(x; t) \stackrel{\text{def}}{=} \inf\{\ell : \bar{L}_f^\ell(x) \leq t \leq \overline{U}_f^\ell(x)\}$$

Outputs can only be closer to the underlying data under this approximate definition which could hurt accuracy, but will still give the desired privacy for inverse sensitivity mechanism. We then extend this definition equivalently for our reflective inverse sensitivity.

**Definition A.3.** For a function $f : \mathcal{X}^n \to \mathbb{R}$ along with $\overline{U}_f^\ell : \mathcal{X}^n \to \mathbb{R}$ and $\bar{L}_f^\ell : \mathcal{X}^n \to \mathbb{R}$, for any $t \in \mathbb{R}$, we let

$$\overline{\mathsf{s}_f}(x; t) \stackrel{\text{def}}{=} \mathrm{sgn}(t - f(x)) \left( \overline{\mathsf{len}_f}(x; t) - \frac{1}{2} \right)$$

We will then be able to show in general that the approximate reflective inverse sensitivity of an output maintains closeness between neighboring datasets. This gives the *approximate asymmetric sensitivity mechanism* for a stream of potential outputs $\{t_i\}$ that calls AboveThreshold and returns $t_k$ when

$$\texttt{AboveThreshold}(x, \{\overline{\mathsf{s}_f}(x; t_i)\}, T = 0) = \{\perp^{k-1}, \top\} \tag{M.3}$$

This will then achieve the same privacy guarantees which is shown in Appendix A.3.

**Theorem A.4.** *Given $f : \mathcal{X}^n \to \mathbb{R}$ and any stream of potential outputs $\{t_i\}$, we have that mechanism M.3 is $(\varepsilon_1 + 2\varepsilon_2)$-DP in general and $(\varepsilon_1 + \varepsilon_2)$-DP if $\overline{\mathsf{s}_f}$ is monotonic.*

## A.2  Analysis for exact asymmetric sensitivity mechanism

We first prove a helper lemma regarding the closeness of the reflective inverse sensitivities for neighboring datasets.

**Lemma A.5.** *Given sample-monotone $f : \mathcal{X}^n \to \mathbb{R}$, we must have $\mathrm{im}(f)$ is a convex set, and for any neighboring datasets $x, x' \in \mathcal{X}^n$ and $t \in \mathrm{im}(f)$ we have $|\mathsf{s}_f(x; t) - \mathsf{s}_f(x'; t)| \leq 1$*

*Proof.* We first show the image is convex. For any $a, b \in \mathrm{im}(f)$, there must be datasets $x_a, x_b \in \mathcal{X}^n$ such that $f(x_a) = a$ and $f(x_b) = b$. Furthermore, by Definition 2.1 we know $d_{\mathsf{ham}}(x_a, x_b) \leq n$, so

$\text{len}_f(\boldsymbol{x}_a; b) \leq n$. Without loss of generality assume $a \leq b$, then by Definition 3.2 for any $c \in [a, b]$ we must have $\text{len}_f(\boldsymbol{x}_a; c) \leq n$ which implies $c \in \text{im}(f)$.

Next we show the closeness between neighboring datasets. First consider the case when $t > \max\{f(\boldsymbol{x}), f(\boldsymbol{x}')\}$ or $t < \min\{f(\boldsymbol{x}), f(\boldsymbol{x}')\}$. This implies $\text{sgn}(t - f(\boldsymbol{x})) = \text{sgn}(t - f(\boldsymbol{x}'))$ and the bound follows from Corollary 2.9. Without loss of generality, assume $f(\boldsymbol{x}) \geq f(\boldsymbol{x}')$ and consider the other case when $f(\boldsymbol{x}) \geq t \geq f(\boldsymbol{x}')$. Due to the fact that they're neighboring $\text{len}_f(\boldsymbol{x}; f(\boldsymbol{x}')) \leq 1$ and $\text{len}_f(\boldsymbol{x}'; f(\boldsymbol{x})) \leq 1$. Applying Definition 3.2, $\text{len}_f(\boldsymbol{x}; t) \leq 1$ and $\text{len}_f(\boldsymbol{x}'; t) \leq 1$ which implies $|s_f(\boldsymbol{x}; t)| \leq 1/2$ and $|s_f(\boldsymbol{x}'; t)| \leq 1/2$ giving the desired bound. $\qquad\square$

With this lemma we can prove Theorem 3.3.

*Proof of Theorem 3.3.* For $t_i \in \text{im}(f)$ we know the sensitivity is at most 1 from Lemma A.5. Further if $t_i \notin \text{im}(f)$ then by the convexity of $\text{im}(f)$ from Lemma A.5 we know that either $t_i < f(\boldsymbol{x})$ and so $s_f(\boldsymbol{x}, t) = -\infty$ for all $\boldsymbol{x} \in \mathcal{X}^n$ or $t_i > f(\boldsymbol{x})$ and so $s_f(\boldsymbol{x}, t) = \infty$ for all $\boldsymbol{x} \in \mathcal{X}^n$. The privacy guarantees then follow from Proposition 2.5. $\qquad\square$

We also provide a proof of Lemma 3.5 connecting the inverse sensitivities to the upper and lower output bounds for sample-monotone functions.

*Proof of Lemma 3.5.* First consider the case when $t < L_f^n(\boldsymbol{x})$ or $t > U_f^n(\boldsymbol{x})$, which implies $\inf\{\ell : L_f^\ell(\boldsymbol{x}) \leq t \leq U_f^\ell(\boldsymbol{x})\} = \infty$ because the infimum of the empty set is infinity. This also implies that $t \notin \text{im}(f)$ so $\text{len}_f(\boldsymbol{x}; t) = \infty$ for the same reason. Next, consider the other case when $L_f^n(\boldsymbol{x}) \leq t \leq U_f^n(\boldsymbol{x})$ and let $k = \inf\{\ell : L_f^\ell(\boldsymbol{x}) \leq t \leq U_f^\ell(\boldsymbol{x})\}$. If there exists $\boldsymbol{x}'$ such that $f(\boldsymbol{x}') = t$ and $d_{\text{ham}}(\boldsymbol{x}, \boldsymbol{x}') = k' < k$, then this would imply $L_f^{k'}(\boldsymbol{x}) \leq t \leq U_f^{k'}(\boldsymbol{x})$, so $\inf\{\ell : L_f^\ell(\boldsymbol{x}) \leq t \leq U_f^\ell(\boldsymbol{x})\} < k$ giving a contradiction. Thus we must have $\text{len}_f(\boldsymbol{x}; t) \geq k$. Furthermore, the sample-monotone definition implies that $\text{len}_f(\boldsymbol{x}; t) \leq k$ because $L_f^k(\boldsymbol{x}) \leq t \leq U_f^k(\boldsymbol{x})$. Therefore, we also have $\text{len}_f(\boldsymbol{x}; t) = k$.

$\qquad\square$

## A.3 Analysis for approximate asymmetric sensitivity mechanism

We again prove a helper lemma regarding the closeness of the approximate inverse sensitivities for neighboring datasets.

**Lemma A.6.** *Given $f : \mathcal{X}^n \to \mathbb{R}$, for any neighboring datasets $\boldsymbol{x}, \boldsymbol{x}' \in \mathcal{X}^n$ and $\inf_x\{f(\boldsymbol{x})\} \leq t \leq \sup_x\{f(\boldsymbol{x})\}$ we have $\left|\overline{\text{len}}_f(\boldsymbol{x}; t) - \overline{\text{len}}_f(\boldsymbol{x}'; t)\right| \leq 1$*

*Proof.* By construction, $L_f^n(\boldsymbol{x}) = \inf_x\{f(\boldsymbol{x})\}$ and $U_f^n(\boldsymbol{x}) = \sup_x\{f(\boldsymbol{x})\}$ because the Hamming distance between any datasets is always at most $n$. Thus we must have $\overline{\text{len}}_f(\boldsymbol{x}; t) \leq n$ and $\overline{\text{len}}_f(\boldsymbol{x}'; t) \leq n$. Without loss of generality, assume $\overline{\text{len}}_f(\boldsymbol{x}; t) \leq \overline{\text{len}}_f(\boldsymbol{x}'; t)$ and $\overline{\text{len}}_f(\boldsymbol{x}; t) = \ell$. By Definition A.1 we then have $\bar{L}_f^{\ell+1}(\boldsymbol{x}') \leq t \leq \bar{U}_f^{\ell+1}(\boldsymbol{x}')$ so $\overline{\text{len}}_f(\boldsymbol{x}'; t) \leq \ell + 1$, which implies our desired inequality.

$\qquad\square$

We then extend this closeness to the approximate reflective inverse sensitivities for neighboring datasets.

**Lemma A.7.** *Given $f : \mathcal{X}^n \to \mathbb{R}$, for any neighboring datasets $\boldsymbol{x}, \boldsymbol{x}' \in \mathcal{X}^n$ and $\inf_x\{f(\boldsymbol{x})\} \leq t \leq \sup_x\{f(\boldsymbol{x})\}$ we have $|\overline{s}_f(\boldsymbol{x}; t) - \overline{s}_f(\boldsymbol{x}'; t)| \leq 1$*

*Proof.* First consider the case when $t > \max\{f(\boldsymbol{x}), f(\boldsymbol{x}')\}$ or $t < \min\{f(\boldsymbol{x}), f(\boldsymbol{x}')\}$. This implies $\text{sgn}(t - f(\boldsymbol{x})) = \text{sgn}(t - f(\boldsymbol{x}'))$ and the bound follows from Lemma A.6. Without loss of generality, assume $f(\boldsymbol{x}) \geq f(\boldsymbol{x}')$ and consider the other case when $f(\boldsymbol{x}) \geq t \geq f(\boldsymbol{x}')$. Due to the fact that they're neighboring and Definition A.2 we have $\overline{\text{len}}_f(\boldsymbol{x}; t) \leq 1$ and $\text{len}_f(\boldsymbol{x}'; t \leq 1$. This implies $|\overline{s}_f(\boldsymbol{x}; t)| \leq 1/2$ and $|\overline{s}_f(\boldsymbol{x}'; t)| \leq 1/2$ giving the desired bound. $\qquad\square$

With these lemmas we can now prove our Theorem A.4.

*Proof of Theorem A.4.* For $t$ such that $\inf_x\{f(x)\} \le t \le \sup_x\{f(x)\}$ we know the sensitivity is at most 1 from Lemma A.7. Otherwise either $t_i < f(x)$ and so $\overline{s_f}(x,t) = -\infty$ for all $x \in \mathcal{X}^n$ or $t_i > f(x)$ and so $\overline{s_f}(x,t) = \infty$ for all $x \in \mathcal{X}^n$. The privacy guarantees then follow from Proposition 2.5. $\qquad\square$

# B Supplemental Results

In this section we provide supplemental results and experiments to our main results. We first show that the asymmetric sensitivity mechanism naturally handles unbounded data for our instantiations with negligible accuracy loss. Next we provide a simple, general, and robust strategy for selecting potential outputs for our method and provide a corresponding empirical study. Finally, we discuss how our methods can also apply to the add-subtract definition of neighboring datasets.

## B.1 Naturally handling unbounded data

As previously discussed in our instantiations from Sections 5 and 6, the functions considered will have infinite upper output bounds if the data is unbounded. Given the iterative nature of the sparse vector technique, we will be able to naturally handle this setting with negligible accuracy loss. In particular, Algorithm 1 outputs the first query above the threshold, and we see from our Definition A.3 that even if $U_f^\ell(x) = \infty$ then we will have that the reflective inverse sensitivity is 1/2 for all possible outputs greater than $f(x)$. Thus each query of potential outputs greater than $f(x)$ is more likely than not to terminate the algorithm. The probability of termination increases even more if the reflective inverse sensitivity is greater than 1/2 but will have minimal effect.

We test this upon our variance instantion by using the bounds from Section 5 and also considering the unbounded case. We also use the same parameters and datasets from Section 5. From Figure 3, we see that the difference in performance for the unbounded setting is slim and thus our method can inherently consider unbounded data.

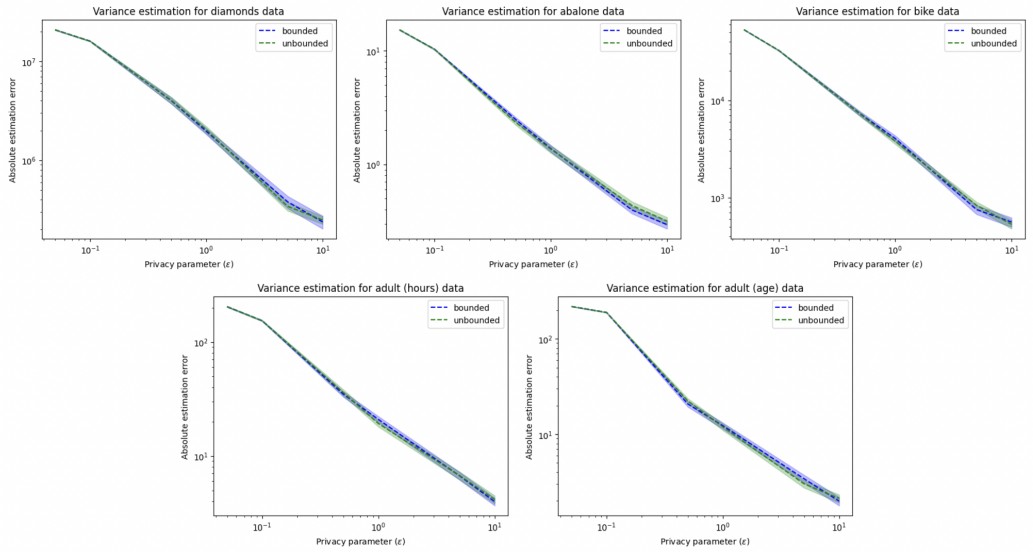

Figure 3: Plots of the absolute error for variance estimation with both reasonable bounds on the data and unbounded data.

While this works for our instantiations, this is aided by each function being non-negative by construction, and applying our method generally to unbounded data will often require an innate lower or upper bound on the function output. In contrast, efficient implementations of the inverse sensitivity mechanism require both upper and lower bounds on the function output, and often further require bounded data for reasonable accuracy such as in our instantiations. More specifically, the inverse sensitivity mechanism uses the upper and lower output bounds from Definition 3.4 to construct intervals and draws an interval from the exponential mechanism and uniformly selects a point from the chosen interval. But this requires setting a data independent upper and lower bound from the function for efficient implementation. This efficient approach was first seen in

an instantiation of a close variant of the inverse sensitivity mechanism for privately computing quantiles Smith (2011).

## B.2   Robust and efficient potential output selection

In order to handle both fully unbounded and partially unbounded functions, we provide an efficient and robust potential output selection method that also borrows from the private quantile literature Durfee (2023), which instantiates a close variant of our asymmetric sensitivity mechanism. In Algorithm 2, we provided the explicit approach for selecting potential outputs when the outputs are non-negative. This same approach can be shifted to handle any other lower bounded functions and symmetrically applied to upper bounded functions. The exponential nature of the potential outputs implies that they will become incredibly large or small within a reasonable number of queries which limits the running time. Specifically, if we set reasonably assume $\beta \geq 1.001$, then $\beta^{50000} > 10^{21}$ and we can simply terminate AboveThreshold after at most 50,000 queries for all datasets without affecting the privacy guarantees.

If the function has no innate bounds then we can call sparse vector technique with two iterations, searching through the positive numbers first, and then searching through the negatives if the first iteration immediately terminated. If the function has both innate upper and lower bounds then we can uniformly select the potential outputs from these bounds.

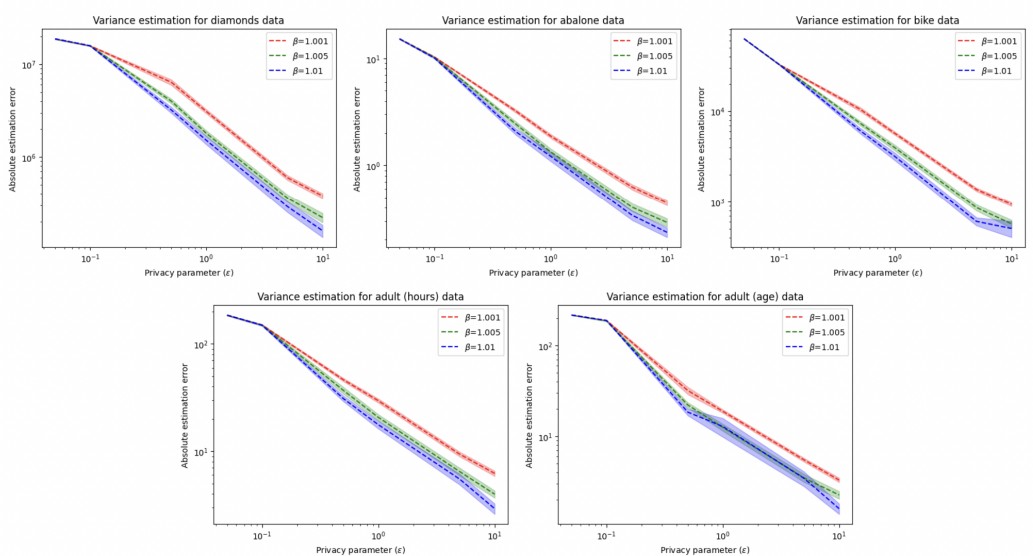

Figure 4: Plot of variance estimation using our method for different $\beta$ parameters

While this methodology introduces a new parameter $\beta$ that must be selected independent of the underlying data, we were able to fix $\beta = 1.005$ as a default and still see high performance across different invocations and datasets. Furthermore, we show here that we could consider other reasonable settings of $\beta$ and still see robustly accurate performance from our methodology. In fact, from our experiments we can see that our empirical results could have been further improved by setting $\beta = 1.01$

## B.3   Add-subtract neighboring

In Definition 2.1, we made the notion of neighboring datasets follow the *swap* definition. Another commonly used notion of neighboring datasets in the differential privacy literature is the *add-subtract* definition where a users data is added or removed between neighbors. Our asymmetric sensitivity mechanism can naturally extend to this notion of Hamming distance as well.

The primary difference would be that all datasets in the data universe would no longer be at most distance $n$ from one another as the size of the dataset could vary. As such, the list of upper and lower output bounds could be infinite, but we can circumvent this with minimal practical impact. Potential outputs far from the underlying data are already incredibly unlikely to be selected so

relaxing the bounds will barely affect the output distribution. As such, we can set the approximate upper and lower bounds to be positive and negative infinity, respectively, which is essentially identical to what was done in Lemma 5.3. To account for this changed definition we would also need to update the upper and lower output bounds for our instantiations.

## C  Intuition and Asymmetric Advantage

In this section, we first give a more visual explanation of our methodology compared to the inverse sensitivity mechanism. We then provide strong intuition upon why our method will substantially improve the estimation accuracy when the sensitivities are asymmetric by more naturally balancing the bias-variance tradeoff. We further supplement this intuition with a formal metric that quantifies the asymmetry of the sensitivities, and we empirically validate that increased asymmetry directly corresponds to improved relative performance of our method. Finally, we give the missing analysis of Lemma 4.1, our theoretical utility guarantees.

### C.1  Visualization of both methods

Recall that the inverse sensitivity method considered the distance metric of any output from the underlying data in Definition 2.8. We then proposed a variant of that definition better suited to applying the sparse vector technique in Definition 3.1.

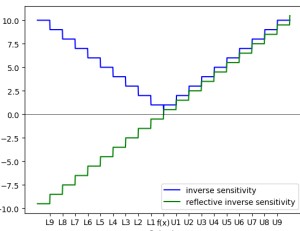

Figure 5:  We provide here an informal visualization of the *inverse sensitivity* and *reflective inverse sensitivity*. For most functions of interest we can just plot these as step functions using the upper and lower output bounds from Definition 3.4. In the plot, we denote $L_f^\kappa(x)$ and $U_f^\kappa(x)$ with $L\kappa$ and $U\kappa$, respectively. We will go into further detail in the next section but note that the sensitivities are perfectly symmetric in this example.

Both of these functions essentially shift between neighboring datasets which allows for maintaining closeness between outputs for each metric. We further note that unlike the *inverse sensitivity*, a shift in the *reflective inverse sensitivity* will be monotonic because of it's increasing nature, allowing for improved privacy guarantees for many instantiations. Given the closeness between outputs for neighboring datasets, we can apply the exponential mechanism or sparse vector technique. Applying our method entails considering an increasing stream of potential output $\{t_i\}$ and (noisily) identifying when the reflective inverse sensitivity crosses from negative to positive by calling the sparse vector technique.

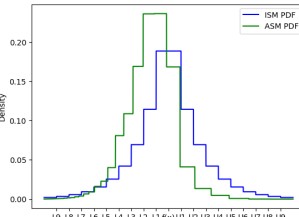

Figure 6:  We provide here an informal visualization of the approximate PDFs for inverse sensitivity mechanism (ISM) and our asymmetric sensitivity mechanism (ASM) when the sensitivities are perfectly symmetric. We slightly alter our mechanism M.3 to uniformly draw an output in $[t_{k-1}, t_k]$ for easier visualization, which implies our PDF will be a step function between the potential outputs.

The iterative nature of the sparse vector technique will most often lead to our method being slightly more likely to find an output less than the true output. This introduces bias into our mechanism but we still remain competitive with inverse sensitivity even in the perfectly symmetric setting.

### C.2  Naturally navigating the bias-variance tradeoff

In Figure 6 we assumed that the sensitivities were perfectly symmetric. More specifically, for $\ell > 0$ we let $\Delta_L^\ell(f; x) \stackrel{\text{def}}{=} L_f^{\ell-1}(x) - L_f^\ell(x)$ and $\Delta_U^\ell(f; x) \stackrel{\text{def}}{=} U_f^\ell(x) - U_f^{\ell-1}(x)$, which are the amount the function can marginally decrease and increase, respectively, by changing the $\ell^{\text{th}}$ individual's

data. Note that for $\ell = 1$ these quantities correspond to the local sensitivity. We then say that the sensitivities are perfectly symmetric if $\Delta_L^\ell(f; \boldsymbol{x}) = \Delta_U^\ell(f; \boldsymbol{x})$ for all $\ell > 0$, which held by construction for our examples in the previous section.

Informally, we will say that the sensitivities are asymmetric if $\Delta_L^\ell(f; \boldsymbol{x}) << \Delta_U^\ell(f; \boldsymbol{x})$ (or the reverse) for most $\ell$ particularly those closer to 0. We now consider an example in which the upper and lower outputs bounds imply asymmetric sensitivities and compare the approximate PDFs for each method.

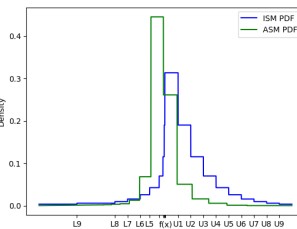

Figure 7: We provide here an informal visualization of the approximate PDFs but with asymmetric sensitivities. We remove the labels for $L_f^1(\boldsymbol{x})$, $L_f^2(\boldsymbol{x})$, and $L_f^3(\boldsymbol{x})$ as they become too condensed around $f(\boldsymbol{x})$ but we keep their tick marks on the x-axis. We again slightly alter our mechanism M.3 to uniformly draw an output in $[t_{k-1}, t_k]$ for easier visualization, which implies our PDF will be a step function between the potential outputs.

For a more encompassing discussion on the bias-variance trade-off, we first consider applying the smooth sensitivity framework to this example. Smooth sensitivity is unbiased in the classical sense (the expected output matches the underlying data) but the noise is added proportional to at least the local sensitivity which would be $\Delta_U^1(f; \boldsymbol{x})$ here. The inverse sensitivity mechanism weakens the unbiased definition to better take advantage of the asymmetry from $\Delta_L^\ell(f; \boldsymbol{x}) << \Delta_U^\ell(f; \boldsymbol{x})$ and we see in Figure 7 that the probability mass of outputs less than $f(\boldsymbol{x})$ becomes much more closely concentrated around $f(\boldsymbol{x})$. However, using their notion of unbiased still limits the extent that it can improve the private estimation. In contrast, a variant of the preprocessing method from Cummings & Durfee (2020) could be applied here and potentially be both unbiased and have low variance if the *a priori* sensitivity parameter closely matches the $\Delta_L^\ell(f; \boldsymbol{x})$ for $\ell$ close to zero. However, applying this same reduced sensitivity parameter, which essentially fixes the variance and must be data independent, would lead to significant bias in Figure 6.

By adding the slight bias towards early stopping from the iterative nature of the sparse vector technique, we can take full advantage of the tighter grouping of the lower output bound to significantly reduce the variance. More specifically, if we map Figure 5 to these lower output bounds, then the reflective inverse sensitivity will be much steeper right below $f(\boldsymbol{x})$, which has the biggest impact upon when sparse vector technique terminates. In fact, we could set the upper output bounds to be infinite and this would only slightly decay the accuracy of our method. This allows us to naturally consider unbounded data for a variety functions, and we specifically examine the effects upon variance estimation in Appendix B.1 with empirical results showing negligible impact upon accuracy. While our method does still have dependence upon the data-independent stream of potential outputs $\{t_i\}$, we provide a simple and general strategy for this selection in Appendix B that robustly maintains accuracy across different invocations and datasets. As a result, the slight bias from our method can utilize the asymmetry for significant improvement while also remaining competitive under perfect symmetry, giving a more inherent optimization of the bias-variance trade-off.

### C.3 Formalizing asymmetry of sensitivities

We previously defined asymmetric sensitivity to informally be a consistent mismatch between $\Delta_L^\ell(f; \boldsymbol{x})$ and $\Delta_U^\ell(f; \boldsymbol{x})$ for most $\ell$. Additionally, this mismatch has the highest impact when $\ell$ is closest to 0, as the outputs farther away from the underlying data are far less likely to be selected. However, the level of privacy further affects this likelihood where smaller $\varepsilon$ implies mismatched $\Delta_L^\ell(f; \boldsymbol{x})$ and $\Delta_U^\ell(f; \boldsymbol{x})$ have a higher impact upon symmetry for larger $\ell$. Therefore, to obtain a formal measurement of asymmetry, we should consider an averaging of $\Delta_L^\ell(f; \boldsymbol{x})$ vs $\Delta_U^\ell(f; \boldsymbol{x})$ over all $\ell$ but with higher weight given to smaller $\ell$ and this weighting should be further scaled by the privacy parameter. We observe that the inverse sensitivity mechanism uniformly draws from $[L_f^\ell, L_f^{\ell-1}]$ and $[U_f^{\ell-1}, U_f^\ell]$ with probability proportional to $\Delta_L^\ell(f; \boldsymbol{x}) \cdot \exp(-\ell \cdot \varepsilon/2)$ and $\Delta^\ell(f; \boldsymbol{x}) \cdot \exp(-\ell \cdot \varepsilon/2)$, respectively, by construction of the exponential mechanism. This then precisely fits with our desired weighting and the probability that the inverse sensitivity mechanism selects an output greater than

$f(x)$ is simply a normalization of $\sum_\ell \Delta_U^\ell(f; x) \cdot \exp(-\ell \cdot \varepsilon/2)$. With this connection of our informal notion to the inverse sensitivity mechanism, we then give a formal definition for measuring the asymmetry of the sensitivities.

**Definition C.1.** Given a function $f : \mathcal{X}^n \to \mathbb{R}$, we measure the asymmetry of the sensitivities by

$$\left| \Pr\left[ M_{\text{inv}}(x) > f(x) \right] - \frac{1}{2} \right|$$

based upon the probability distribution from M.1.

Accordingly, we have that the sensitivities are symmetric if $f(x)$ is the median of the inverse sensitivity mechanism. But this property does not imply accurate estimation as the variance could still be quite large. However, it is not surprising that it will perform relatively worse than our method for more asymmetric sensitivities. We empirically test this conjecture across different levels of sensitivity symmetry, which we also vary by toggling the level of privacy.

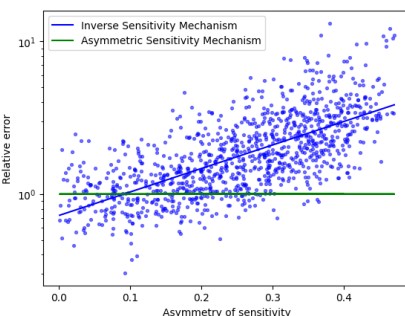

Figure 8: We consider a range of randomly sampled output bounds and privacy parameters and compute the absolute error of our asymmetric sensitivity mechanism (ASM) and the inverse sensitivity mechanism (ISM) over a small number of random draws. We plot the (error ISM) / (error ASM) corresponding to the asymmetry of the sensitivities for the given output bounds and privacy parameter.

For our simulations we uniformly distribute the lower and upper output bounds and we toggle the range of the upper output bounds to vary the level of asymmetry. We also consider upper output bounds that are more heavy tailed and toggle the level of privacy, thereby determining the impact of the tail, to vary the asymmetry of the sensitivities. For each simulated output bounds and privacy parameter, we compute the asymmetry of the sensitivities and we make several calls to both methods for private estimations and compute the average absolute error of each. While these simulations are not all-encompassing, they empirically validate our provided intuition, and we will further see in our instantiations of functions with inherent asymmetry that our methodology gives substantially improved estimates.

### C.4 Analysis of theoretical utility guarantees

In this section we provide the proof of Lemma 4.1. Our proof will first bound the probability that we output a value that's too small and then bound the probability that we output a value that's too large. As a result, we provide two helper lemmas for each direction that are close variants of Theorem 3.24 in Dwork et al. (2014).

**Lemma C.2.** *For any sequence of $m$ queries $f_1, ..., f_m$ with sensitivity 1 such that $|\{i \leq m : f_i(x) \geq T - \alpha\}| = 0$, then Algorithm 1 (where we set $\varepsilon_1 = \varepsilon_2$ and $\varepsilon = \varepsilon_1 + 2\varepsilon_2$) will terminate during these queries with probability at most $m \cdot e^{-\alpha\varepsilon/3}$*

*Proof.* We know that exponential noise is non-negative, so to terminate at query $i$, the noisy result must exceed the threshold. By the CDF of the exponential distribution and our assumption, we have $\Pr_{\nu_i \sim \text{Expo}(3/\varepsilon)}[f_i(x) + \nu_i > T] \leq e^{-\alpha\varepsilon/3}$. Applying a union bound over all $m$ queries gives our desired result. $\qquad\square$

**Lemma C.3.** *For any sequence of $m$ queries $f_j, ..., f_{j+m}$ with sensitivity 1 such that $|\{i \in (j, j+m) : f_i(x) < T + \alpha\}| = 0$, then Algorithm 1 (where we set $\varepsilon_1 = \varepsilon_2$ and $\varepsilon = \varepsilon_1 + 2\varepsilon_2$) will terminate at query $j + m$ or later with probability at most $e^{-\alpha\varepsilon/3}/m$*

*Proof.* In order to terminate at query $j + m$ or later we need that the noisy threshold is above all previous noisy queries. Let $v_T \sim \mathsf{Expo}(3/\varepsilon)$ and $v_i \sim \mathsf{Expo}(3/\varepsilon)$. There are $m-1$ indices in $(j, j+m)$ so by independence we have

$$\frac{1}{m} = \Pr\left[v_T > \max_{i \in (j, j+m)} v_i\right]$$

and furthermore by change of variable and the PDF of the exponential distribution

$$\Pr\left[v_T > \max_{i \in (j, j+m)} v_i\right] = e^{\alpha\varepsilon/3}\Pr\left[v_T - \alpha > \max_{i \in (j, j+m)} v_i\right]$$

Due to our assumption that $f_i(x) \geq T + \alpha$ for all $i \in (j, j+m)$, we see that $\Pr\left[v_T - \alpha > \max_{i \in (j, j+m)} v_i\right]$ is an upper bound on the probability of still continuing at step $j + m$. This implies our desired claim.

$\square$

*Proof of Lemma 4.1.* We first show $\Pr\left[M(x) < L^{\ln k}(x)\right] < C/ke^{\varepsilon/6}$ for a constant $C$. By Definition 3.1 and Lemma 3.5 we know that for any $t_i < L^{\ln k+1}(x)$ we have $\mathsf{s}_f(x; t_i) < T - \ln k$. Furthermore, we assumed that $f(x) \leq \beta^C$ for a constant $C$ and $t_i = \beta^i - 1$ so $t_C > f(x) \geq L^{\ln k+1}(x)$. We then apply Lemma C.2 with $\alpha = \ln k$ and $m = C$, which implies $\Pr\left[M(x) < L^{\ln k}(x)\right] < C/ke^{\varepsilon/6}$ for a constant $C$ as desired.

Next, we want to show that $\Pr\left[M(x) > \beta^k(f(x) + 1)\right] < C/ke^{\varepsilon/6}$ for a constant $C$. Using the fact that $f(x)+1 \geq 1$ there are at least $k-1$ values of $t_i$ such that $t_i \in [f(x), \beta^k(f(x)+1)]$. By Definition 3.1 and Lemma 3.5 we know that for any $t_i \in [f(x), \beta^k(f(x) + 1)]$ we have $\mathsf{s}_f(x; t_i) \geq T + 1/2$. We then apply Lemma C.3 with $\alpha = 1/2$ and $m = k$ to get that $\Pr\left[M(x) > \beta^k(f(x) + 1)\right] \leq 1/ke^{\varepsilon/6}$ as desired.

Combining these inequalities gives our desired

$$\Pr\left[|M(x) - f(x)| < \max\{|f(x) - L^{\log k+1}(x)|, (\beta^k - 1)(f(x) + 1)\}\right] > 1 - O\left(\frac{1}{ke^{\varepsilon/6}}\right)$$

$\square$

**Corollary C.4.** *Let $M$ denote the mechanism from M.3 with $t_i = \beta^i - 1$ where $\beta > 1$ and we let $\varepsilon = \varepsilon_1 + 2\varepsilon_2$ with $\varepsilon_1 = \varepsilon_2$ in the call to Algorithm 1. Given non-negative $f : \mathcal{X} \to \mathbb{R}$ we have*

$$Pr\left[|M(x) - f(x)| < \max\{|f(x) - L^{\log k+1}(x)|, (\beta^k - 1)(f(x) + 1)\}\right] > 1 - O\left(\frac{1}{ke^{\varepsilon/6}}\right)$$

*when $f(x) \leq \beta^{O(1)}$.*

*Proof.* The proof follows identically but instead of applying Lemma 3.5 we can directly use Definition A.2 and note that setting $L_f^\ell \equiv \bar{L}_f^\ell$ satisfies Definition A.1 $\square$

# D   Additional Analysis of Variance Invocation

In this section we provide the necessary proofs for the efficient variance instantiation of our method. Most of the analysis could be considered folklore properties of variance as it is such a well-studied statistical property, but we duplicate some of these properties for completeness. We first provide a known alternative definition of variance.

**Definition D.1.** Let $\mathcal{X} = \mathbb{R}$ and for $x \in \mathcal{X}^n$, we can equivalently define variance as

$$\mathbf{Var}\,[x] \stackrel{\text{def}}{=} \frac{1}{n^2}\sum_{i=1}^n\sum_{j=1}^n\frac{1}{2}(x_i - x_j)^2$$

We will also add some helpful notation where for any $S \subseteq [n]$ we let $x_S = \{x_i : i \in S\}$ and similarly for any $1 \le a \le b \le n$ we let $x_{[a:b]} = (x_a, ..., x_b)$.

## D.1 Analysis for Lemma 5.2

We first prove a helper lemma that if $\ell$ individuals data is changed in order to minimize variance, then those values should be set to be the mean of the remaining values.

**Lemma D.2.** *For any $x \in \mathcal{X}^n$ and any $S \subset [n]$, we have that*

$$\inf_{x_S} \mathbf{Var}\left[x\right] = \frac{n - |S|}{n} \mathbf{Var}\left[x_{[n] \setminus S}\right]$$

*Proof.* By least squares minimization we know that $\sum_{i \in [n] \setminus S}(x_i - y)^2$ is minimized by setting

$$y = \frac{1}{n - |S|} \sum_{i \in [n] \setminus S} x_i \overset{\text{def}}{=} \bar{x}_{[n] \setminus S}$$

If we set $x_i = \bar{x}_{[n] \setminus S}$ for all $i \in S$, then we have $\bar{x} = \bar{x}_{[n] \setminus S}$ and we have $\sum_{i \in S}(x_i - \bar{x})^2 = 0$ which must be minimal by the non-negativity of squared error. Combining these properties we get

$$\inf_{x_S} \mathbf{Var}\left[x\right] = \frac{1}{n} \sum_{i \in [n] \setminus S} (x_i - \bar{x}_{[n] \setminus S})^2$$

We can then apply our Definition 5.1 to get our desired result.

$\square$

With this helper lemma we can then provide the proof of our lower output bounds through a proof by contradiction.

*Proof of Lemma 5.2.* By Lemma D.2 we have that

$$L_{\mathbf{Var}}^{\ell}(x) = \min_{S \subset [n] : |S| = \ell} \frac{n - \ell}{n} \mathbf{Var}\left[x_{[n] \setminus S}\right]$$

We will then give our desired claim through a proof by contradiction. Suppose this is minimized by $S \subset [n]$ with some $i \in S$ such that there exists $j, k \notin S$ where $x_j < x_i < x_k$. Let $S_j = (S \setminus i) \cup j$ and $S_k = (S \setminus i) \cup k$. To obtain our contradiction it then suffices to show that

$$\min\left\{ \mathbf{Var}\left[x_{[n] \setminus S_j}\right], \mathbf{Var}\left[x_{[n] \setminus S_k}\right] \right\} < \mathbf{Var}\left[x_{[n] \setminus S}\right]$$

Applying our alternative formulation of variance from Definition D.1 we have

$$\mathbf{Var}\left[x_{[n] \setminus S}\right] = \frac{1}{n - \ell} \sum_{a \in [n] \setminus S} \sum_{b \in [n] \setminus S} \frac{1}{2}(x_a - x_b)^2$$

Through cancellation of like terms we have

$$\mathbf{Var}\left[x_{[n] \setminus S_j}\right] < \mathbf{Var}\left[x_{[n] \setminus S}\right] \iff \sum_{a \in [n] \setminus (S_j \cup i)} (x_a - x_i)^2 < \sum_{a \in [n] \setminus (S \cup j)} (x_a - x_j)^2$$

By construction we have $(S_j \cup i) = (S \cup j)$, so by the convexity of least squares minimization and the fact that $x_j < x_i$, we have that this inequality holds if $x_i \le \bar{x}_{[n] \setminus (S \cup j)}$. Equivalently, we have

$$x_i \ge \bar{x}_{[n] \setminus (S \cup k)} \Rightarrow \mathbf{Var}\left[x_{[n] \setminus S_k}\right] < \mathbf{Var}\left[x_{[n] \setminus S}\right]$$

Further, we know that $\bar{x}_{[n]\setminus(S\cup j)} > \bar{x}_{[n]\setminus(S\cup k)}$ because $x_j < x_k$. Therefore we must have either $x_i \leq \bar{x}_{[n]\setminus(S\cup j)}$ or $x_i \geq \bar{x}_{[n]\setminus(S\cup k)}$ which implies

$$\min\left\{ \mathbf{Var}\left[x_{[n]\setminus S_j}\right], \mathbf{Var}\left[x_{[n]\setminus S_k}\right] \right\} < \mathbf{Var}\left[x_{[n]\setminus S}\right]$$

and gives our desired contradiction.

$\square$

## D.2 Analysis for Lemma 5.3

In this section we prove that the approximate lower output bounds fit Definition A.1 and can be efficiently computed. We note that these lower output bounds will still maintain high accuracy because the bounds further away from the underlying data have far less effect upon the mechanism and accordingly, loose bounds will have little effect.

*Proof of Lemma 5.3.* Recall that we assumed a fixed constant, $c$ and defined $\bar{L}^\ell_{\mathbf{Var}}(x) = L^\ell_{\mathbf{Var}}(x)$ for $\ell \leq c$ and $\bar{L}^\ell_{\mathbf{Var}}(x) = 0$ for $\ell > c$. By construction, we know that variance is non-negative, so we must have $L^\ell_{\mathbf{Var}}(x) \geq 0$ for all $\ell$. This then implies $\bar{L}^\ell_f(x) \leq L^\ell_f(x)$ for all $\ell$. Furthermore, if $\ell > c$ then we immediately have $\bar{L}^\ell_f(x) \geq \bar{L}^{\ell+1}_f(x')$. Otherwise $L^\ell_f(x) = \bar{L}^\ell_f(x)$ and we know by definition that $L^\ell_f(x) \geq L^{\ell+1}_f(x')$ which implies $\bar{L}^\ell_f(x) \geq \bar{L}^{\ell+1}_f(x')$

For the runtime, by Lemma 5.2 we see that it suffices to compute $\mathbf{Var}\left[x_{[\ell+1-i:n-i]}\right]$ for all $0 \leq i \leq \ell \leq c$, where we assume $x_1 \leq ... \leq x_n$. However, this ordering only matters the largest and smallest $c$ values, so we compute these in $O(n + c\log(c))$ time. Further, we compute $\sum_{i=1}^n x_i$ and $\sum_{i=1}^n x_i^2$ in $O(n)$ time and will utilize the well-known fact that variance can be computed in $O(1)$ time with these quantities. We can then just iterate through all $i, \ell$ such that $0 \leq i \leq \ell \leq c$ updating the sum of variables and squares to compute each $\mathbf{Var}\left[x_{[\ell+1-i:n-i]}\right]$ in $O(1)$ time taking a total of $O(c^2)$ time.

$\square$

## D.3 Analysis for Lemma 5.4

*Proof of Lemma 5.4.* We first show that for any neighboring datasets $x, x'$ we have $\bar{U}^\ell_{\mathbf{Var}}(x) \leq \bar{U}^{\ell+1}_{\mathbf{Var}}(x')$. By construction, this reduces to showing that $\mathbf{Var}[x] \leq \mathbf{Var}[x'] + (b-a)^2/n$. Let $i$ be the index such that $x_i \neq x'_i$. Applying the alternative variance formulation in Definition D.1 and cancelling like terms reduces this to showing

$$\frac{1}{n^2}\sum_{j\neq i}(x_i - x_j)^2 \leq \frac{1}{n^2}\sum_{j\neq i}(x'_i - x_j)^2 + \frac{(b-a)^2}{n}$$

We assumed that all datasets were restricted to $[a,b]^n$ so we must then have $\sum_{j\neq i}(x_i - x_j)^2 \leq n(b-a)^2$, which then implies our desired inequality by the non-negativity of squared error.

Next we show that $U^\ell_{\mathbf{Var}}(x) \leq \bar{U}^\ell_{\mathbf{Var}}(x)$. It suffices to show that for an arbitrary $x'$ such that $d_{\mathsf{ham}}(x, x') = \ell$ we have $\mathbf{Var}[x'] \leq \mathbf{Var}[x] + \frac{\ell(b-a)^2}{n}$. We previously showed that $\mathbf{Var}[x] \geq \mathbf{Var}[x'] + (b-a)^2/n$ for any $x, x'$ such that $d_{\mathsf{ham}}(x, x') = 1$, so our claim then follows inductively.

Therefore, our construction of $\bar{U}^\ell_{\mathbf{Var}}$ satisfies Definition A.1 as desired.

$\square$

# E  Additional Analysis of Model Evaluation Invocations

In this section, we provide the definitions and implementation details for applying our method to machine learning model evaluation functions. We further unify this analysis by considering applying our methodology to linearly separable functions.

We now define our datasets as $(\boldsymbol{x}, y) \in \mathcal{X}^n \times \mathbb{R}^n$ and we set $d_{\mathsf{ham}}((\boldsymbol{x}, y), (\boldsymbol{x}', y')) = |\{i : \boldsymbol{x}_i \neq \boldsymbol{x}'_i \text{ or } y_i \neq y'_i\}|$ to be the Hamming distance between datasets. We will be considering binary classification and multi-class classification, so we define cross-entropy loss for both.

**Definition E.1.** Given machine learning model for binary-classification $\omega : \mathcal{X} \rightarrow \mathbb{R}$ and assume all $y_i \in \{0, 1\}$, let

$$\mathsf{BCE}_\omega(\boldsymbol{x}, y) = \sum_{i=1}^n -y_i \log\left(\frac{1}{1 + e^{-\omega(\boldsymbol{x}_i)}}\right) - (1 - y_i) \log\left(\frac{e^{-\omega(\boldsymbol{x}_i)}}{1 + e^{-\omega(\boldsymbol{x}_i)}}\right)$$

Similarly, given machine learning model for multi-classification $\omega : \mathcal{X} \rightarrow \mathbb{R}^c$ with $c$ classes and assume all $y_i \in [c]$, let

$$\mathsf{CE}_\omega(\boldsymbol{x}, y) = -\sum_{i=1}^n \log\left(\frac{e^{\omega(\boldsymbol{x}_i)_{y_i}}}{\sum_{j=1}^c e^{\omega(\boldsymbol{x}_i)_j}}\right)$$

We also define the mean squared error and mean absolute error.

**Definition E.2.** Given a dataset $(\boldsymbol{x}, y)$ and machine learning model $\omega : \mathcal{X} \rightarrow \mathbb{R}$, we define

$$\mathsf{MSE}_\omega(\boldsymbol{x}, y) = \frac{\sum_{i=1}^n (\omega(\boldsymbol{x}_i) - y_i)^2}{n}$$

and

$$\mathsf{MAE}_\omega(\boldsymbol{x}, y) = \frac{\sum_{i=1}^n |\omega(\boldsymbol{x}_i) - y_i|}{n}$$

### E.1 Application to linearly separable functions

In this section we consider all *linearly separable functions* $f : \mathcal{X}^n \rightarrow \mathbb{R}$ such that

$$f(\boldsymbol{x}) = \sum_{i=1}^n \mathcal{L}(\boldsymbol{x}_i)$$

where $\mathcal{L} : \mathcal{X} \rightarrow \mathbb{R}$. We could also extend our results here to $\mathcal{L}$ that are specific to the index, but for simplicity we will restrict our consideration. Without loss of generality assume the indices are ordered such that $\mathcal{L}(\boldsymbol{x}_i) \leq \mathcal{L}(\boldsymbol{x}_{i+1})$. We first provide lower output bounds for these functions.

**Lemma E.3.** *Given a linearly separable function $f : \mathcal{X} \rightarrow \mathbb{R}$, then*

$$L_f^\ell(\boldsymbol{x}) = \sum_{i=1}^{n-\ell} \mathcal{L}(\boldsymbol{x}_i) + \ell \cdot \inf_{\boldsymbol{x}_k \in \mathcal{X}} \{\mathcal{L}(\boldsymbol{x}_k)\}$$

*Proof.* Changing any individual's data can only change their contribution to the sum by the linearly separable property. Therefore, decreasing the $\ell$ individuals with the highest contribution to the sum must minimize the function for datasets with Hamming distance $\ell$ from the underlying data. $\square$

Similarly, we provide upper output bounds for linearly separable functions.

**Lemma E.4.** *Given a linearly separable function $f : \mathcal{X} \rightarrow \mathbb{R}$, then*

$$U_f^\ell(\boldsymbol{x}) = \sum_{i=\ell}^n \mathcal{L}(\boldsymbol{x}_i) + \ell \cdot \sup_{\boldsymbol{x}_k \in \mathcal{X}} \{\mathcal{L}(\boldsymbol{x}_k)\}$$

*Proof.* Follows equivalently to the proof of Lemma E.3 $\square$

We then provide approximate relaxations of these bounds that will allow for easier application.

**Lemma E.5.** *Given a linearly separable function* $f : \mathcal{X} \to \mathbb{R}$, *then approximate upper and lower sensitivity bounding functions* $\overline{U}_f^\ell : \mathcal{X}^n \to \mathbb{R}$ *and* $\overline{L}_f^\ell : \mathcal{X}^n \to \mathbb{R}$ *satisfy Definition A.1 for*

$$\overline{L}_f^\ell(\boldsymbol{x}) = \sum_{i=1}^{n-\ell} \mathcal{L}(\boldsymbol{x}_i) + \ell \cdot a \qquad \text{and} \qquad \overline{U}_f^\ell(\boldsymbol{x}) = \sum_{i=\ell}^{n} \mathcal{L}(\boldsymbol{x}_i) + \ell \cdot b$$

*if* $a \leq \inf_{\boldsymbol{x}_k \in \mathcal{X}}\{\mathcal{L}(\boldsymbol{x}_k)\}$ *and* $b \geq \sup_{\boldsymbol{x}_k \in \mathcal{X}}\{\mathcal{L}(\boldsymbol{x}_k)\}$

*Proof.* We show this the approximate lower output bounds and the upper follow equivalently. We first have $\overline{L}_f^\ell(\boldsymbol{x}) \leq L_f^\ell(\boldsymbol{x})$ by construction. Next, for neighboring $\boldsymbol{x}, \boldsymbol{x}'$, let $j$ be the index at which $\boldsymbol{x}_j \neq \boldsymbol{x}'_j$. We assumed an ordering to the indices for simplicity, but we equivalently have

$$\overline{L}_f^\ell(\boldsymbol{x}) = \min_{S \subseteq [n]\,:\,|S|=n-\ell} \Big\{ \sum_{i \in S} \mathcal{L}(\boldsymbol{x}_i) \Big\} + \ell \cdot a$$

For some given $\ell$, let $S_{\boldsymbol{x}}$ be the subset of indices that minimizes this for $\overline{L}_f^\ell(\boldsymbol{x})$. If $j \notin S_{\boldsymbol{x}}$ then we have $\overline{L}_f^\ell(\boldsymbol{x}) \geq \overline{L}_f^\ell(\boldsymbol{x}')$, so $\overline{L}_f^\ell(\boldsymbol{x}) \geq \overline{L}_f^{\ell+1}(\boldsymbol{x}')$. Otherwise, we know that

$$\overline{L}_f^{\ell+1}(\boldsymbol{x}') = \min_{S \subseteq [n]\,:\,|S|=n-(\ell+1)} \Big\{ \sum_{i \in S} \mathcal{L}(\boldsymbol{x}'_i) \Big\} + (\ell+1) \cdot a \leq \sum_{i \in S_{\boldsymbol{x}} \setminus j} \mathcal{L}(\boldsymbol{x}_i) + (\ell+1) \cdot a$$

and because $a \leq \mathcal{L}(\boldsymbol{x}_j)$ then this implies $\overline{L}_f^\ell(\boldsymbol{x}) \geq \overline{L}_f^{\ell+1}(\boldsymbol{x}')$.

$\square$

We further show that our approximate bounds allow for monotonic reflective inverse sensitivities which implies improved privacy guarantees.

**Lemma E.6.** *Given a linearly separable function* $f : \mathcal{X} \to \mathbb{R}$, *along with approximate upper and lower sensitivity bounding functions* $\overline{U}_f^\ell : \mathcal{X}^n \to \mathbb{R}$ *and* $\overline{L}_f^\ell : \mathcal{X}^n \to \mathbb{R}$ *such that*

$$\overline{L}_f^\ell(\boldsymbol{x}) = \sum_{i=1}^{n-\ell} \mathcal{L}(\boldsymbol{x}_i) + \ell \cdot a \qquad \text{and} \qquad \overline{U}_f^\ell(\boldsymbol{x}) = \sum_{i=\ell}^{n} \mathcal{L}(\boldsymbol{x}_i) + \ell \cdot b$$

*where* $a \leq \inf_{\boldsymbol{x}_k \in \mathcal{X}}\{\mathcal{L}(\boldsymbol{x}_k)\}$ *and* $b \geq \sup_{\boldsymbol{x}_k \in \mathcal{X}}\{\mathcal{L}(\boldsymbol{x}_k)\}$. *For any neighboring datasets* $\boldsymbol{x}, \boldsymbol{x}'$ *we have that either* $\overline{s}_f(\boldsymbol{x}; t) \leq \overline{s}_f(\boldsymbol{x}'; t)$ *for all* $t \in \mathbb{R}$ *or* $\overline{s}_f(\boldsymbol{x}; t) \geq \overline{s}_f(\boldsymbol{x}'; t)$ *for all* $t \in \mathbb{R}$

*Proof.* Without loss of generality, assume $f(\boldsymbol{x}) \leq f(\boldsymbol{x}')$ and we will show $\overline{s}_f(\boldsymbol{x}; t) \geq \overline{s}_f(\boldsymbol{x}'; t)$ for all $t \in R$.

We first consider $t \in [f(\boldsymbol{x}), f(\boldsymbol{x}')]$. Given that the datasets are neighboring, we must have $L_f^1(\boldsymbol{x}) \leq t \leq U_f^1(\boldsymbol{x})$ and $L_f^1(\boldsymbol{x}') \leq t \leq U_f^1(\boldsymbol{x}')$. By Definition A.1 and Definition A.2, we have that $\overline{\text{len}}_f(\boldsymbol{x}; t) \leq 1$ and $\overline{\text{len}}_f(\boldsymbol{x}'; t) \leq 1$. Given that $t \in [f(\boldsymbol{x}), f(\boldsymbol{x}')]$ this then implies $\overline{s}_f(\boldsymbol{x}; t) \in \{1/2, 0\}$ and $\overline{s}_f(\boldsymbol{x}'; t) \in \{-1/2, 0\}$. Therefore $\overline{s}_f(\boldsymbol{x}; t) \geq \overline{s}_f(\boldsymbol{x}'; t)$.

Next consider the case when $t < f(\boldsymbol{x})$. This implies $\text{sgn}(t - f(\boldsymbol{x})) = \text{sgn}(t - f(\boldsymbol{x}')) = -1$ and also that $\overline{\text{len}}_f(\boldsymbol{x}; t) \geq 1$ and $\overline{\text{len}}_f(\boldsymbol{x}'; t) \geq 1$. By Lemma E.7 we have that $\overline{L}_f^\ell(\boldsymbol{x}) \leq \overline{L}_f^\ell(\boldsymbol{x}')$ for all $\ell \geq 0$, which implies $\overline{\text{len}}_f(\boldsymbol{x}; t) \leq \overline{\text{len}}_f(\boldsymbol{x}'; t)$ and therefore $\overline{s}_f(\boldsymbol{x}; t) \geq \overline{s}_f(\boldsymbol{x}'; t)$.

Finally consider $t > f(\boldsymbol{x}')$. This implies $\text{sgn}(t - f(\boldsymbol{x})) = \text{sgn}(t - f(\boldsymbol{x}')) = 1$ and also that $\overline{\text{len}}_f(\boldsymbol{x}; t) \geq 1$ and $\overline{\text{len}}_f(\boldsymbol{x}'; t) \geq 1$. By Lemma E.7 we have that $\overline{U}_f^\ell(\boldsymbol{x}) \leq \overline{U}_f^\ell(\boldsymbol{x}')$ for all $\ell \geq 0$, which implies $\overline{\text{len}}_f(\boldsymbol{x}; t) \geq \overline{\text{len}}_f(\boldsymbol{x}'; t)$ and therefore $\overline{s}_f(\boldsymbol{x}; t) \geq \overline{s}_f(\boldsymbol{x}'; t)$.

$\square$

We will also require the following helper lemma in order prove Lemma E.6.

**Lemma E.7.** *Given a linearly separable function $f : \mathcal{X} \to \mathbb{R}$, along with approximate upper and lower sensitivity bounding functions $\overline{U}_f^\ell : \mathcal{X}^n \to \mathbb{R}$ and $\overline{L}_f^\ell : \mathcal{X}^n \to \mathbb{R}$ such that*

$$\overline{L}_f^\ell(\boldsymbol{x}) = \sum_{i=1}^{n-\ell} \mathcal{L}(\boldsymbol{x}_i) + \ell \cdot a \qquad \text{and} \qquad \overline{U}_f^\ell(\boldsymbol{x}) = \sum_{i=\ell}^{n} \mathcal{L}(\boldsymbol{x}_i) + \ell \cdot b$$

*where $a \leq \inf_{\boldsymbol{x}_k \in \mathcal{X}}\{\mathcal{L}(\boldsymbol{x}_k)\}$ and $b \geq \sup_{\boldsymbol{x}_k \in \mathcal{X}}\{\mathcal{L}(\boldsymbol{x}_k)\}$. For any neighboring datasets $\boldsymbol{x}, \boldsymbol{x}'$, if $f(\boldsymbol{x}) \leq f(\boldsymbol{x}')$ then $\overline{L}_f^\ell(\boldsymbol{x}) \leq \overline{L}_f^\ell(\boldsymbol{x}')$ and $\overline{U}_f^\ell(\boldsymbol{x}) \leq \overline{U}_f^\ell(\boldsymbol{x}')$ for all $\ell \geq 0$*

*Proof.* Let $j$ be the index at which $\boldsymbol{x}_j \neq \boldsymbol{x}'_j$, which implies $\mathcal{L}(\boldsymbol{x}_j) \leq \mathcal{L}(\boldsymbol{x}'_j)$ because of our linearly separable property. We assumed an ordering to the indices for simplicity, but we equivalently have

$$\overline{L}_f^\ell(\boldsymbol{x}) = \min_{S \subseteq [n] : |S| = n-\ell} \Big\{ \sum_{i \in S} \mathcal{L}(\boldsymbol{x}_i) \Big\} + \ell \cdot a$$

For a given $\ell$ let $S_{\boldsymbol{x}'}$ denote the set of indices that minimizes $\overline{L}_f^\ell(\boldsymbol{x}')$. If $j \in S_{\boldsymbol{x}'}$ then we have

$$\overline{L}_f^\ell(\boldsymbol{x}') = \mathcal{L}(\boldsymbol{x}'_j) + \sum_{i \in S_{\boldsymbol{x}'} \setminus j} \mathcal{L}(\boldsymbol{x}_i) + \ell \cdot a \geq \mathcal{L}(\boldsymbol{x}_j) + \sum_{i \in S_{\boldsymbol{x}'} \setminus j} \mathcal{L}(\boldsymbol{x}_i) + \ell \cdot a = \sum_{i \in S_{\boldsymbol{x}'}} \mathcal{L}(\boldsymbol{x}_i) + \ell \cdot a \geq \overline{L}_f^\ell(\boldsymbol{x})$$

Similarly, if $j \notin S_{\boldsymbol{x}'}$ then

$$\overline{L}_f^\ell(\boldsymbol{x}') = \sum_{i \in S_{\boldsymbol{x}'}} \mathcal{L}(\boldsymbol{x}_i) + \ell \cdot a \geq \overline{L}_f^\ell(\boldsymbol{x})$$

The proof for $\overline{U}_f^\ell(\boldsymbol{x}) \leq \overline{U}_f^\ell(\boldsymbol{x}')$ follows equivalently.

$\square$

### E.2 Efficient cross-entropy loss instantiation

We will provide the efficient instantiation for multi-class cross entropy loss as this can easily be extended to binary cross entropy loss. Without loss of generality, assume the indices are ordered such that

$$-\log\left(\frac{e^{\omega(\boldsymbol{x}_i)_{y_i}}}{\sum_{j=1}^{c} e^{\omega(\boldsymbol{x}_i)_j}}\right) \leq -\log\left(\frac{e^{\omega(\boldsymbol{x}_{i+1})_{y_{i+1}}}}{\sum_{j=1}^{c} e^{\omega(\boldsymbol{x}_{i+1})_j}}\right)$$

We can then provide the lower output bounds for cross entropy loss

**Lemma E.8.** *Given a dataset $(\boldsymbol{x}, y)$ and machine learning model $\omega : \mathcal{X} \to \mathbb{R}^c$, then*

$$\overline{L}_{\mathsf{CE}_\omega}^\ell(\boldsymbol{x}, y) = -\sum_{i=1}^{n-\ell} \log\left(\frac{e^{\omega(\boldsymbol{x}_i)_{y_i}}}{\sum_{j=1}^{c} e^{\omega(\boldsymbol{x}_i)_j}}\right)$$

*Proof.* We apply Lemma E.3 and Lemma E.5 and we observe that

$$\inf_{\boldsymbol{x}_i, y_i}\left\{ -\log\left(\frac{e^{\omega(\boldsymbol{x}_i)_{y_i}}}{\sum_{j=1}^{c} e^{\omega(\boldsymbol{x}_i)_j}}\right) \right\} \geq 0$$

$\square$

As seen with variance in Section 5, we have that $U_{\mathsf{CE}_\omega}^1(\boldsymbol{x}, y) = \infty$ which implies that cross-entropy loss has inherently asymmetric sensitivities. Similarly, we will need to restrict the range of these values in order to apply inverse sensitivity mechanism even though our method could easily handle the unbounded setting. We provide a proof for these upper output bounds in Appendix E.

**Lemma E.9.** *Given a dataset $(\boldsymbol{x}, y)$ and machine learning model $\omega : \mathcal{X} \to \mathbb{R}^c$ where we restrict $\omega(\boldsymbol{x}_i) \in [a, b]^c$ for all i, then*

$$\overline{U}^{\ell}_{\mathrm{CE}_{\omega}}(\boldsymbol{x}, y) = -\left( \ell \cdot \log\left( \frac{e^{a-b}}{e^{a-b} + c - 1} \right) + \sum_{i=\ell+1}^{n} \log\left( \frac{e^{\omega(\boldsymbol{x}_i)_{y_i}}}{\sum_{j=1}^{c} e^{\omega(\boldsymbol{x}_i)_j}} \right) \right)$$

*Proof.* We apply Lemma E.4 and Lemma E.5, and we observe that with our restricted bounds we have

$$\sup_{\boldsymbol{x}_i, y_i}\left\{ -\log\left( \frac{e^{\omega(\boldsymbol{x}_i)_{y_i}}}{\sum_{j=1}^{c} e^{\omega(\boldsymbol{x}_i)_j}} \right) \right\} \le -\log\left( \frac{e^{a-b}}{e^{a-b} + c - 1} \right)$$

$\square$

The corresponding algorithm for instantiating cross-entropy loss with our method is similar to Algorithm 2. The privacy guarantees from Theorem 5.5 also follow equivalently, but we can additionally improve the privacy to be $(\varepsilon_1 + \varepsilon_2)$-DP by applying Lemma E.6 to achieve monotonicity. We can also achieve $O(n\log(n) + q)$ runtime by computing all of our approximate upper and lower bounds, but we could also easily employ the same strategy of setting these bounds to be infinity and zero, respectively, for all $\ell > c$ where we set $c = 100$. This then gives the linear runtime, where we also utilize the fact that we will never run AboveThreshold from more than 50,000 queries.

## E.3 Efficient implementation for regression evaluation

We will only provide the efficient implementation for MSE in this section as MAE will follow identically. Without loss of generality, assume the indices are ordered such that $(\omega(\boldsymbol{x}_i) - y_i)^2 \le (\omega(\boldsymbol{x}_{i+1}) - y_{i+1})^2$.

**Lemma E.10.** *Given a dataset $(\boldsymbol{x}, y)$ and machine learning model $\omega : \mathcal{X} \to \mathbb{R}$, then*

$$\overline{L}^{\ell}_{\mathrm{MSE}_{\omega}}(\boldsymbol{x}, y) = \frac{\sum_{i=1}^{n-\ell}(\omega(\boldsymbol{x}_i) - y_i)^2}{n}$$

*Proof.* We apply Lemma E.3 and Lemma E.5, and we observe that

$$\inf_{\boldsymbol{x}_i, y_i}\{\omega(\boldsymbol{x}_i) - y_i)^2\} \ge 0$$

$\square$

As seen with variance in Section 5, we have that $U^1_{\mathrm{MSE}_{\omega}}(\boldsymbol{x}, y) = \infty$ which implies that MSE has inherently asymmetric sensitivities. Similarly, we will need to restrict the range of these values in order to apply inverse sensitivity mechanism even though our method could easily handle the unbounded setting.

**Lemma E.11.** *Given a dataset $(\boldsymbol{x}, y)$ and machine learning model $\omega : \mathcal{X} \to \mathbb{R}$ where we restrict $\omega(\boldsymbol{x}_i) \in [a, b]$ and $y_i \in [a, b]$ for all i, then*

$$\overline{U}^{\ell}_{\mathrm{MSE}_{\omega}}(\boldsymbol{x}, y) = \frac{\ell(b-a)^2 + \sum_{i=\ell+1}^{n}(\omega(\boldsymbol{x}_i) - y_i)^2}{n}$$

*Proof.* We apply Lemma E.4 and Lemma E.5, and we observe that for our bounded setting

$$\sup_{\boldsymbol{x}_i, y_i}\{\omega(\boldsymbol{x}_i) - y_i)^2\} \le (b-a)^2$$

$\square$

The corresponding algorithm for instantiating MSE with our method is identical to Algorithm 2. The privacy guarantees from Theorem 5.5 also follow equivalently, but we can additionally improve the privacy to be $(\varepsilon_1 + \varepsilon_2)$-DP by applying Lemma E.6 to achieve monotonicity. We can also achieve $O(n \log(n) + q)$ runtime by computing all of our approximate upper and lower bounds, but we could also easily employ the same strategy of setting these bounds to be infinity and zero, respectively, for all $\ell > c$ where we set $c = 100$. This then gives the linear runtime, where we also utilize the fact that we will never run AboveThreshold from more than 50,000 queries.

