# OpenReview forum: "Instance-Specific Asymmetric Sensitivity in Differential Privacy"
_NeurIPS.cc/2024/Conference — NeurIPS 2024 poster_

### Official Review · Reviewer_v2LJ · 2024-07-11

**Soundness:** 3
**Presentation:** 3
**Contribution:** 3
**Rating:** 7
**Confidence:** 4

**Summary:**

In this paper, authors propose a new instance specific sensitivity based method for differentially private queries. In differentially private literature, sensitivity of the query (or the function of interest) is an important quantity, which affects the amount of noise we need to add in order to guarantee DP. In majority of DP mechanisms, the sensitivity is considered as the maximum change of the query when a single data point is changed in the data. Scaling the noise level with this global sensitivity might lead to large amount of noise added which then negatively affects the utility of the method. Instead of using this global sensitivity, another line of work is to use so called smooth sensitivity which calibrates the noise based on the data set at hand, leading to lower level of perturbation and better utility. In this paper, authors extend this line of work with their new sensitivity definition called *reflective inverse sensitivity* and the corresponding release method *asymmetric sensitivity mechanism* which is based on earlier AboveThreshold mechanism. Authors prove privacy and utility guarantees for the proposed method, and empirically demonstrate its benefits over the earlier smooth sensitivity based methods providing up to a magnitude difference in the accuracy.

**Strengths:**

The sensitivity of a function is typically hard to bound without some strong prior knowledge on the private data set. Hence majority of the DP works resort to some ad-hoc bounds, e.g. through clipping of the function or the data, to limit the sensitivity of the function. This can create bias to the output, and analyzing the bias before the application is difficult. Hence the proposed method, which does not suffer from this limitation, is interesting. While the proposed solution builds on top of existing literature, the novelty of this works contribution clearly separates them from prior literature.

The theoretical analysis seems sound to me, and the utility result is an important finding. However, I think the exposition of this result could be further improved and will come back to this in the weaknesses.

I also believe that the proposed mechanism has broad application potential. For example tuning classifiers trained with public data based on their accuracy on the private data. The experiments in such tasks demonstrate significant benefits over the prior works.

**Weaknesses:**

In general the paper reads well, but it took me a while to parse the main story together. I would suggest authors to simplify some parts of the introduction to make the paper easier to approach. For example, I was rather confused by the use of "inverse of a function" in the introduction, since I don't think it is clear at this point what kind of functions the paper talks about. Therefore it was unclear to me what would be the argument of the function to take the inverse from. This becomes very clear later in the paper, but perhaps some motivating example. to introduce reader to the topic of inverse sensitivity, would be beneficial.

I also think the utility result, Lemma 4.1, could be better discussed. There is some further discussion about the result in the appendix, but I would suggest moving some of that to the main paper. It would be very important to at least discuss, what is the key take-away of the utility guarantee. Otherwise it is somewhat hard to appreciate it.

**Questions:**

- Can you clarify what is the motivation for subsampling 1000 samples from each data set in your variance experiments? Is the full data variance task too easy to show any difference between the methods? Or does the O(n) runtime become a bottleneck for running the experiments on the full data?

- typos
    * "... in it’s full generality ..."
    * in the end of page 6.: "even if $U_1^k(x) = \infty$", I guess it should be $U_f^1(x)$?

**Limitations:**

I believe the limitations are properly discussed.

---

> ### Author Rebuttal · Authors · 2024-08-06
>
> We thank the reviewer for their thorough review of our work and incredibly helpful feedback. In retrospect, we are disappointed with our decision to relegate much of the intuition to the appendix in deference to including more of the concrete results and will be sure to better incorporate aspects of this discussion into the main body.
>
>
> $\textbf{Questions:}$
>
>
> The decision to subsample 1000 samples was entirely following from previous DP papers we had read that did this method for their empirical studies. Their justification was that this bootstrapping better simulates new draws from the underlying distribution of the original data. As a result, it better ensures that the proposed method does not happen to only perform well on that specific dataset, but performs well generally on datasets drawn from the underlying distribution. The specific amount of 1000 was likely just a result of being a "clean" number (power of 10) and less than the total data size for most datasets. There are no computational difficulties with using larger datasets (within reason).
>
>
> Thanks for catching the typos! The reviewer is absolutely right in their correction of the second typo (our apologies for the confusion).

---

> > ### Comment · Reviewer_v2LJ · 2024-08-12
> >
> > Thank you for your response! I'm happy to keep my score as is.

---

### Official Review · Reviewer_55Ug · 2024-07-12

**Soundness:** 2
**Presentation:** 2
**Contribution:** 2
**Rating:** 6
**Confidence:** 4

**Summary:**

This paper proposes a new instance specific asymmetric sensitivity under differential privacy by building on the well-studied inverse sensitivity mechanism which adapts to the hardness of the local data according to the inverse closeness to the underlying ground dataset. It also develops some theoretical guarantee and performs empirical validation.

**Strengths:**

The paper deals with an important question in differential privacy.

**Weaknesses:**

(1) The paper is not well-written and is very hard to follow. The abstract and introduction mentions the notion of asymmetric sensitivity without any intuitive explanation. Due to the lack of this important intuition, it is very difficult to understand well the contributions of this paper although the authors list their works in Section 1.3.
(2)The important definition of asymmetric sensitivity mechanism (Line 187) is very unclear. The benefit of asymmetrcity compared with the (symmetric) inverse sensitivity mechanisms in (Asi and Duchi 2020.
(3)The paper fails to provide the motivations and explanations for theoretical results in a clear and convincing way. The authors consider many concepts and notions but don’t provide their clear logical connection to the main topic of the paper.
(4)The citations usually appear in the main text not in the footnote (like in Page 7).

**Questions:**

(1)Line 208: Why the computation time is O(log n?
(2)In Definition 3.1,  why 1/2? Can we replace it with 0 or 1?
(3)Could you please rewrite the definition of asymmetric sensitivity mechanisms in the form of sparse vector technique or in the form of DP mechanisms? What is the benefit of asymmeticity? Could you provide a concrete example where the sensitivity is asymmetric and explain how your techniques can be applied to it?
(5)Use the above example to illustrate the variance-bias tradeoff.
(6)What is the advantage of asymmetric sensitivity mechanisms in this paper over inverse sensitivity mechanisms in the literature?

**Limitations:**

yes

---

> ### Author Rebuttal · Authors · 2024-08-06
>
> We know that the reviewing burden can be significant and are very sorry to have added to this burden by not presenting our results more clearly. It's clear that we attempted to fit in too many of our results in the main body and neglected better explaining the intuition of our methodology. In particular, we relegated our main intuition section to the appendix (Section C) and are very disappointed in ourselves for this decision in retrospect.
>
>
> $\textbf{Questions:}$
>
>
> We greatly apologize for not better clarifying these questions the reviewer raises. We attempt to do so $\textit{concisely}$ here, but are more than happy to give further detail.
>
>
> $\textit{Question 1 response.}$ The array $L^n_f(x),...,L^1_f(x), f(x), U^1_f(x),...,U^n_f(x)$ is ordered due to construction of Definition 3.4, and so inserting $t$ into this ordered array takes $O(\log n)$ time. Where $t$ lands in this array then immediately implies the value of $len(x;t)$ by Lemma 3.5. For example, if $L^i_f(x) < t < L^{i-1}_f(x)$ then $len(x;t) = i$.
>
>
> $\textit{Question 2 response.}$ This is best understood through Figure 5 where the reflective inverse sensitivity is a step-function and each step must be at most distance 1 apart to give the privacy guarantees (Lemma A.5). Subtracting 1/2 ensures this desired property (we're happy to give further detail).
>
>
> $\textit{Question 3 (a) response.}$ We apologize for not clarifying, but sparse vector technique essentially just iteratively calls AboveThreshold (see section 3.6 of The Algorithmic Foundations of Differential Privacy). Given that we only need one call, we directly called AboveThreshold (which is in our preliminaries), but our mechanism is then essentially already written in the form of a simple call to sparse vector technique.  Intuitively, if we consider Figure 5 again, our mechanism considers outputs in increasing order and tries to identify when the reflective inverse sensitivity crosses from negative to positive, ie exceeds the threshold of 0 which occurs at f(x), by applying sparse vector technique.
>
>
> $\textit{Question 3 (b) response.}$ We view the benefit in terms of the relative advantage of our method (answered in question 6)
>
>
> $\textit{Question 3 (c) response.}$ To be clear, our technique can be generally applied and is just more effective than other methods when the sensitivities are asymmetric. An example of asymmetric sensitivities is for variance. If we change one individual's data point the amount the variance decreases is bounded, but the amount it increases can be infinite (section D.1 has full details).
>
>
> $\textit{Question 6 response.}$ The benefit of our method comes when the sensitivities are asymmetric (changing an individual's data increases the function more than decreasing it). To best understand this we plot the PDF of both for perfectly symmetric sensitivities in Figure 6. As you can see, our method biases the output to be slightly below the true value. We then plot the PDF with asymmetric sensitivities in Figure 7 and see that our method has more probability mass around the true value in this setting. This intuition is discussed further in Section C and validated empirically with our instantiations in section 5 and 6, and theoretically shown in section 4.
>
>
> $\textit{Question 5 response.}$ Our method is biased towards selecting outputs below the true value (discussed in question 6) and so the sensitivity of decreasing the function has more impact than the sensitivity of increasing the function on the variance of our estimate. Connecting to example in 3(c), for the unbiased estimators, smooth sensitivity and inverse sensitivity, the noise added is infinite (so infinite variance) because the local sensitivity is infinite. In contrast, adding some bias allows us to have variance that is proportional to the sensitivity of decreasing the function (which is bounded) and we still achieve high utility (for example, see Figure 3)

---

> > ### Comment · Reviewer_55Ug · 2024-08-13
> > **I will rasie my evaluation**
> >
> > Thank you for your rebuttal.   I am satisfied with the answers. I will raise my evaluation to 6.

---

### Official Review · Reviewer_J6Z2 · 2024-07-12

**Soundness:** 3
**Presentation:** 3
**Contribution:** 3
**Rating:** 7
**Confidence:** 3

**Summary:**

The paper introduces a new notion of instance-dependent sensitivity, asymmetric sensitivity, to release general function queries. Compared with inverse sensitivity, it can better capture the underlying instance’s asymmetry, i.e. when changing a data point causing the function value to increase or decrease at different magnitudes. The authors also design an implementation framework for asymmetric sensitivity based private query release using the sparse vector mechanism. Proofs of privacy and utility guarantees as well as empirical evaluation are provided to support the advantage of this algorithm.

**Strengths:**

* The proposed asymmetric sensitivity and the query-releasing framework built upon it are novel and will benefit future research.
* The paper is well-written, providing clear intuition of the methods, theoretical guarantees, and empirical evaluation.

**Weaknesses:**

As the author noted in the paper (lines 178-179), the proposed method may not yield significant improvements when applied to vector-valued queries due to the structure of high-dimensional space.

typo: L^{\log k}} instead of L_f^{\log k}} in Lemma 4.1 and its proof.

**Questions:**

From empirical evaluation (Figures 1, 2, and 3), the improvement of the asymmetric sensitivity mechanism upon the inverse sensitivity mechanism seems to be higher in the high privacy regime than in the low privacy regime. Can the authors comment on this?

**Limitations:**

The authors provide this part in the checklist

---

> ### Author Rebuttal · Authors · 2024-08-05
>
> We thank the reviewer for their thorough review of our work, helpful feedback, and catching an important typo.
>
>
> $\textbf{Question:}$
>
> This is a phenomenal question and a great catch by the reviewer. To answer it really requires digging deep into the intuition upon why our method performs better with asymmetric sensitivities. A brief summary is that:
>
> 1. Greater asymmetry of the underlying instance correlates to better relative performance of our method (see Figure 8) and explicitly quantifying asymmetry not only depends on the underlying instance but also the privacy parameter (Section C.3 has full details)
>
> 2. For variance and model evaluations, we generally have that the asymmetry of the underlying instance increases for higher privacy parameters.
>
>
> $\textbf{Longer explanation of 1. (Section C.3 has full details):}$
>
>
> Recall in line 234 that we informally consider the sensitivities to be asymmetric if $|f(x) - U_f^k(x)| >> |f(x) - L_f^k(x)| $ for most $k$ (or vice versa), which is to say that changing an individual's data can generally increase the function more than decrease it. In order to explicitly quantify this we need to average this magnitude of difference over all $k$. However, this averaging should not be uniform because the mechanisms are more likely to select outputs closer to $f(x)$, i.e. within $[L_f^k(x), L_f^{k-1}(x)]$ or $[U_f^{k-1}(x), U_f^k(x)]$ for smaller $k$ values (this is a slight oversimplification). Essentially, the more likely the mechanism is to select outputs closer to $f(x)$, the more important the magnitude of difference is for small $k$. The privacy parameter directly affects this likelihood, so for higher privacy parameters the magnitude of difference for larger $k$ is more impactful in the quantification of asymmetry. We empirically confirm this intuition in our experiments for Figure 8.
>
>
>
>
> $\textbf{Longer explanation of 2:}$
>
>
> The summary here is that for variance and model evaluations we generally have that the magnitude of difference between $|f(x) - U_f^k(x)|$ and $|f(x) - L_f^k(x)| $ becomes larger as $k$ increases. As such, for higher privacy parameters the asymmetry of sensitivities increases and our method sees increased relative advantage.
>
>
> To understand this we'll restrict to the variance instantiation for simplicity and assume the data is Gaussian. Changing one individual's data to minimize variance takes the min or max data point and moves it to the mean, and similarly to maximize the variance takes the data point closest to the mean and moves it to the boundary (note this is a slight oversimplification). This generalizes to changing $k$ individual's data. Due to the bell curve shape of the Gaussian, there are far fewer outliers compared to points close to the mean. As such, the amount we can decrease the variance lessens much more quickly with respect to $k$. This then results in the magnitude of difference between $|f(x) - U_f^k(x)|$ and $|f(x) - L_f^k(x)| $ becoming larger as $k$ increases.
>
>
> While the real data we use is not necessarily Gaussian, we still generally expect it to be somewhat concentrated as opposed to uniformly spread and the same notion holds. This same idea applies to model evaluation where we expect more errors to be closer to zero as opposed to very large.

---

> > ### Comment · Reviewer_J6Z2 · 2024-08-12
> > **Official comment by reviewer J6Z2**
> >
> > Thank you for answering my question! I will keep my score.

---

### Official Review · Reviewer_Zw6m · 2024-07-12

**Soundness:** 3
**Presentation:** 3
**Contribution:** 2
**Rating:** 5
**Confidence:** 2

**Summary:**

The paper proposes an asymmetric sensitivity mechanism for the private estimation of functions with asymmetric outputs, such as variance. This mechanism combines the inverse sensitivity mechanism with the sparse vector technique to handle asymmetric sensitivities effectively. The proposed method is efficient and demonstrates superior performance in variance estimation and model evaluation tasks (e.g., for regression and classification models) compared to existing methods like inverse sensitivity mechanisms and smooth sensitivity mechanisms.

**Strengths:**

- This paper utilises the asymmetric sensitivity property (e.g. "changing an individual's data can generally increase the function more than decrease it"). This property holds for many problems.
- The proposed method is of practical relevance and has efficient implementations for many applications.

**Weaknesses:**

The theoretical guarantee can be vacuous in some reasonable settings. For example, the upper bound in lemma 4.1 can be loose when $f(x)\ll 1$ for all $x$.

**Questions:**

Is there a general recipe for efficiently implementing the asymmetric sensitivity mechanism beyond the specific examples provided (i.e., variance estimation and model evaluations)?

**Limitations:**

See weaknesses.

---

> ### Author Rebuttal · Authors · 2024-08-05
>
> We thank the reviewer for their thorough review of our work and their helpful feedback.
>
>
> $\textbf{Weaknesses:}$
>
>
> Could the reviewer please explain further why they would classify our bounds as "vacuous" in reasonable settings? We certainly agree with the reviewer that asymptotic bounds can sometimes be loose in practice, but we feel this applies to the previous work as well. For the scenario the reviewer brings up, $f(x) << 1$, our upper bound essentially becomes absolute error (of approximately $\beta^k - 1$) instead of relative error, but this is still non-trivial. Additionally, it is still true in this setting that our bounds are superior to previous work as $U_f^1(x) \rightarrow \infty$.
>
>
>
>
> $\textbf{Question:}$
>
>
> As was originally pointed out in the smooth sensitivity paper, unfortunately there are functions where computing even just local sensitivity is computationally infeasible, which is required for exact smooth sensitivity, inverse sensitivity, as well as our asymmetric sensitivity. Computing efficient and meaningful approximations of local sensitivity is then still highly dependent upon the function of interest for all of these methods.

---

### Official Review · Reviewer_GRDP · 2024-07-15

**Soundness:** 2
**Presentation:** 3
**Contribution:** 2
**Rating:** 5
**Confidence:** 4

**Summary:**

This paper provides a new algorithmic framework for differentially privately computing general functions that adapts to the "local" sensitivity of the underlying dataset. It follows previous work's paradigm which is to sample outcomes with probabilities exponentially decreasing in how "far away" the current dataset is from one that produces that particular outcome. The mechanism works by finding the reflective inverse sensitivity which is slightly different from the conventionally used inverse sensitivity; then it tests a series of monotone outcomes and outputs the first one whose reflective inverse sensitivity passes the threshold of 0. The result is a potentially biased mechanism hence the known performance guarantees for related mechanisms such as inverse sensitivity mechanism do not directly apply. Since the evaluation of the reflective inverse sensitivity can be inefficient, the paper then discusses practical tactics for approximating it. The mechanism is carried out on the private variance problem and the experiments show promising results where the new mechanism beats previous ones with a significant lead.

**Strengths:**

The mechanism used in the paper is original in that the defined variant of inverse sensitivity is asymmetric, and it does improve the performance in the case study of private variance estimation. The paper is organized well, the writing rather clear. The mechanism could be a preferrable alternative to the state-of-the-art, its practical use mostly depending on whether we can easily find approximate functions for the asymmetric inverse sensitivity that are efficient to compute but that can be said of most exponential mechanisms.

**Weaknesses:**

The greatest weakness of this paper is perhaps the lack of justifications of why the proposed mechanism should work better; especially the asymmetric part. What it does is skip the t's smaller than f(x) and only look at bigger ones, hence we are moving the mass probability from one side of the distribution and put it on the other. The paper provides some intuition but I still fail to see why it should help. I am inclined to believe that in the private variance computation it helps because we know variances cannot be negative.

By and large I think it is pointing out a phenomenon noticeable in practice, and I don't doubt that it could be applicable to more cases than have already been observed. Still there is the missing piece of puzzle of why we are observing this and how to extend this knowledge to other related problems. The community could benefit greatly from that missing theoretical analysis and it will complete the story.

**Questions:**

How does the performance of this algorithm compare to, say, running propose-test-release by testing for a series of fixed number of t's and test $\{len_f(x;t)\}$ by comparing them one by one to some threshold (maybe 1/2) as used here and return the first t that passes this test? It seems close to this mechanism in nature, since mostly it's just aiming at picking the smallest t to get a positive $len_f(X;t)$; further the result is also sort of like an exponential mechanism. Do the authors have any intuition of this? It is perhaps a symmetric version of this.

**Limitations:**

The authors have adequately addressed them.

---

> ### Author Rebuttal · Authors · 2024-08-05
>
> We thank the reviewer for their thorough review of our work and their helpful feedback.
>
>
> $\textbf{Weaknesses:}$
>
>
> We greatly appreciate the reviewer's desire for intuition and strongly agree. Unfortunately we decided to put our main intuition section in the appendix (Section C) due to space considerations and are disappointed in our decision in retrospect. It is unreasonable to ask the reviewer to read this section, so we will briefly discuss here how it addresses the reviewer's concerns.
>
>
> First, the bias in the PDF from our method is actually towards selecting outputs below f(x) and not above as the reviewer seems to suggest (see Figure 6).  As a result, when the sensitivities are asymmetric then the bias in the PDF is beneficial for our method (see Figure 7). We explicitly define this asymmetry (Definition C.1) and then empirically test this more generally (Figure 8) to show that higher asymmetry of sensitivity clearly correlates with better relative performance of our method, and this is theoretically supported by our Lemma 4.1
>
>
> Connecting this to one of our use cases, variance, the reviewer is absolutely correct that non-negativity contributes to improved performance but the root cause is that changing one individual's data will most often increase variance more than decrease it (section D.1 has full details). In particular, if the data is unbounded then one individual can infinitely increase the variance by changing their data. This is also true of our other use cases (section E has full details) and in general we expect this to be the case for functions with inherent lower bounds (like non-negativity as the reviewer suggests) or inherent upper bounds.
>
>
> We're more than happy to explain further and provide additional pointers if it would help.
>
>
> $\textbf{Question:}$
>
>
> This is a great question and is definitely a close variant of our approach. But it does seem that the process the reviewer is suggesting (continual querying until a threshold is exceeded) matches more with SVT / AboveThreshold as opposed to using propose-test-release which only tests one value and releases if the test passes.
>
>
> This type of method using SVT would work and we agree with the reviewer that it's kind of a symmetric version of our method. But our intuition is then that this would only potentially be more effective with symmetric sensitivities and would also still be worse than inverse sensitivity in that setting.

---

### Decision · Program_Chairs · 2024-09-25

**Decision:**

Accept (poster)

**Comment:**

This paper proposes a new framework for differentially private computation of functions. The reviewers were all positive about the paper. While the reviewers pointed out some shortcomings of the exposition, overall these seem like minor revisions that can be addressed by the authors.